

# Identifying Chemical Aerosol Signatures using Optical Suborbital Observations: How much can optical properties tell us about aerosol composition?

Meloë S. F. Kacenelenbogen[1], Qian Tan[1,2], Sharon P. Burton[3], Otto P. Hasekamp[4], Karl D. Froyd[5], Yohei Shinozuka[1,2], Andreas J. Beyersdorf[6], Luke Ziemba[3], Kenneth L. Thornhill[3], Jack E. Dibb[7], Taylor Shingler[8], Armin Sorooshian[8], Reed W. Espinosa[9,10], Vanderlei Martins[10], Jose L. Jimenez[5], Pedro Campuzano-Jost[5], Joshua P. Schwarz[11], Matthew S. Johnson[1], Jens Redemann[12], Gregory L. Schuster[3]

[1]NASA Ames Research Center, Moffett Field, CA, 94035, USA
[2]Bay Area Environmental Research Institute (BAERI), Moffett Field, CA, 94035, USA
[3]NASA Langley Research Center, Hampton, VA, 23666, USA
[4]SRON, Netherlands Institute for Space Research, Utrecht, 3584, Netherlands
[5]Cooperative Institute for Research in Environmental Sciences (CIRES), University of Colorado, Boulder, CO, 80309, USA
[6]California State University, San Bernardino (CSUSB), San Bernardino, CA, 92407, USA
[7]University of New Hampshire, Durham, NH, 03824, USA
[8]University of Arizona, Tucson, AZ, 85721, USA
[9]NASA Goddard Space Flight Center, Greenbelt, MD, 20771, USA
[10]University of Maryland Baltimore County (UMBC), Baltimore, MD 21250, USA
[11]Chemical Sciences Division, NOAA Earth System Research Laboratory, Boulder, CO, USA
[12]University of Oklahoma, Norman, OK 73019, USA

*Correspondence to*: Meloë S.F. Kacenelenbogen (meloe.s.kacenelenbogen@nasa.gov)

**Abstract.** Improvements in air quality and Earth's climate predictions require improvements of the
aerosol speciation in chemical transport models, using observational constraints. Aerosol speciation (e.g., organic aerosols, black carbon, sulfate, nitrate, ammonium, dust or sea salt) is typically determined using *in situ* instrumentation. Continuous, routine surface network aerosol composition measurements are not uniformly widespread over the globe. Satellites, on the other hand, can provide a maximum coverage of the horizontal and vertical atmosphere but observe aerosol optical properties (and not aerosol speciation)
based on remote sensing instrumentation. Combinations of satellite-derived aerosol optical properties can inform on air mass aerosol types (AMTs e.g., clean marine, dust, polluted continental). However, these AMTs are subjectively defined, might often be misclassified and are hard to relate to the critical parameters that need to be refined in models.
In this paper, we derive AMTs that are more directly related to sources and hence to speciation. They are
defined, characterized, and derived using simultaneous *in situ* gas-phase, chemical and optical instruments on the same aircraft during the Study of Emissions and Atmospheric Composition, Clouds, and Climate Coupling by Regional Surveys (SEAC[4]RS, US, summer of 2013). First, we prescribe well-informed AMTs that display distinct aerosol chemical and optical signatures to act as a training AMT





dataset. These *in situ* observations reduce the errors and ambiguities in the selection of the AMT training
dataset. We also investigate the relative skill of various combinations of aerosol optical properties to
define AMTs and how much these optical properties can capture dominant aerosol speciation.

We find distinct optical signatures for biomass burning (from agricultural or wildfires), biogenic and dust-
influence AMTs. Useful aerosol optical properties to characterize these signatures are the extinction
angstrom exponent (EAE), the single scattering albedo, the difference of single scattering albedo in two
wavelengths, the absorption coefficient, the absorption angstrom exponent (AAE), and the real part of the
refractive index (RRI). We find that all four AMTs studied when prescribed using mostly airborne *in situ*
gas measurements, can be successfully extracted from at least three combinations of airborne *in situ*
aerosol optical properties (e.g., EAE, AAE and RRI) over the US during SEAC$^4$RS. However, we find
that the optically based classifications for BB from agricultural fires and polluted dust include a large
percentage of misclassifications that limit the usefulness of results relating to those classes.

The technique and results presented in this study are suitable to develop a representative, robust and
diverse source-based AMT database. This database could then be used for widespread retrievals of AMTs
using existing and future remote sensing suborbital instruments/networks. Ultimately, it has the potential
to provide a much broader observational aerosol data set to evaluate chemical transport and air quality
models than is currently available by direct *in situ* measurements.

This study illustrates how essential it is to explore existing airborne datasets to bridge chemical and optical
signatures of different AMTs, before the implementation of future spaceborne missions (e.g., the next
generation of Earth Observing System (EOS) satellites addressing Aerosol, Cloud, Convection and
Precipitation (ACCP) designated observables).


## 1 Introduction

Aerosols have an important yet uncertain impact on the Earth's radiation budget (e.g., Boucher et al.,
2013) and human health (e.g., US EPA, 2011, 2016; Lim et al., 2012; Lanzi, 2016; Landrigan et al., 2018;
Wu et al., 2020). In particular, aerosols impact human health by increasing the number of cases of
emphysemas, lung cancers, diabetes, hypertensions and premature deaths (e.g., Wichmann et al., 2000;
Pope et al., 2002; Lim et al., 2012; Lelieveld et al., 2019, 2015; Stirnberg et al., 2020; Nault et al., 2021);
this particularly holds true for specific species of aerosols with high oxidative potential (e.g., Daellenbach
et al. 2020).

Chemical Transport Models (CTMs) and General Circulations Models (GCMs) derive aerosol optical
properties and estimate the Radiative Forcing due to aerosol-radiation interactions (RFari), based on
simulated water uptake, simulated aerosol mass concentrations, simplified aerosol size distributions and
assumed aerosol refractive indices per species (Chin et al., 2002). RFari for individual aerosol species
(e.g., sulfate, black carbon (BC), organic aerosol (OA, typically classified into primary, POA, and
secondary organic aerosol, SOA), nitrate, biomass burning (BB)) are less certain than the total RFari
(Boucher et al., 2013; Myhre et al., 2013). Myhre et al. (2013) present a large AeroCom Phase II inter-
model spread in the Radiative Forcing (RF) of several aerosol species. BC, for example, had a 40%
relative standard deviation in RFari. Inter-model diversity in estimates of RFari is caused in part by



different methods for estimating aerosol properties (e.g., emissions, transport, chemistry, deposition, optical properties (Loeb and Su, 2010)), and to a lesser extent by surface and cloud albedos, water vapor absorption, and radiative transfer schemes (e.g., Randles et al., 2013; Myhre et al., 2013; Stier at al., 2013; Thorsen et al., 2021).

In order to constrain model simulations, data assimilation techniques have been adopted using optimal estimation methods and observational constraints that we separate in four main groups. The first group of constraints consists in column-integrated aerosol optical properties from passive orbital and/or suborbital instruments (e.g., Collins et al., 2001; Yu et al., 2003; Generoso et al., 2007; Adhikary et al., 2008; Niu et al., 2008; Zhang et al., 2008; Benedetti et al., 2009; Schutgens et al., 2010; Kumar et al., 2019; Tsikerdekis et al., 2021). The second group consists in fine aerosol mass concentrations from airborne and/or ground-based instruments (e.g., Lin et al., 2008; Pagowski and Grell, 2012). The third group consists in a combination of *in situ* gas-phase measurements (e.g., $SO_2$, $NO_2$, $O_3$, CO), fine aerosol mass concentrations from ground-based instruments and column-integrated aerosol optical properties from passive orbital instruments (e.g., Ma et al., 2019). The fourth group consists in surface (e.g., Kahnert, 2008, Yumimoto et al., 2008; Uno et al., 2008) and space-based aerosol lidar profiles (e.g., Sekiyama et al., 2010; Zhang et al., 2011), which are used to constrain aerosol mass and extinction. Constraining model-predicted aerosol mass concentrations with passive satellite total column-integrated aerosol properties has been shown to be useful to constrain model-predicted AOD. This is the case for the single-channel visible Aerosol Optical Depth (AOD) retrievals from the Moderate Resolution Imaging Spectroradiometer (MODIS) sensor (e.g., Yu et al., 2003; Zhang et al., 2008; Benedetti et al., 2009; Sessions et al., 2015; Buchard et al., 2017; Kumar et al., 2019; Ma et al., 2019). However, this process does not correct the uncertainty associated with the simulated vertical distribution of aerosols, nor can it derive aerosol chemical speciation. On the other hand, assimilation of satellite-derived optical properties related to particle size (e.g., Extinction Angstrom Exponent, EAE) and light absorption (e.g., Single Scattering Albedo, SSA) represents a step forward (e.g., Tsikerdekis et al., 2021). Another way to improve estimates of speciated RFari would be to use satellite-derived total column speciated aerosol mass concentration to adjust the mass concentration of individual aerosol masses when applying data assimilation techniques in the model (and potentially the emission/chemistry/transport processes driving them). However, currently no satellite-derived retrievals of aerosol chemical speciation exist.

Let us note an important distinction between what is called aerosol speciation and aerosol type. On the one hand, we define aerosol speciation the inherent chemical composition of the aerosol (also called aerosol component), the chemical species that are represented in CTMs (e.g., BC, OA, brown carbon, sulfate, nitrate, ammonium, dust, sea salt). These are typically defined to match the operational quantities reported by in-situ instruments. On the other hand, the Air Mass Aerosol Type (AMT) is representative of typical aerosol mixes associated with certain seasons and geographical locations. It is a coarse definition (qualitative) of the aerosol size, shape and color that dominates an air mass (e.g., clean marine, dust, polluted continental, clean continental, polluted dust, smoke, and stratospheric in the case of CALIOP/ CALIPSO, Cloud-Aerosol Lidar with Orthogonal Polarization/ Cloud-Aerosol Lidar and Infrared Pathfinder Satellite Observation (Omar et al., 2009)).

On the one hand, recent techniques infer aerosol speciation from A-Train's POLDER (Polarization and Directionality of Earth's Reflectances) passive satellite observations on board the PARASOL platform


using an inverse modeling framework, which consists in fitting satellite observations to model estimates by adjusting aerosol emissions. POLDER measures polarized radiances in 14-16 viewing directions at 443, 670 and 865 nm and retrieves aerosol optical properties over land (Deuzé et al., 2001) and over ocean (Herman et al., 2005) using its standard retrieval algorithm. In addition, two alternate POLDER retrieval algorithms from the SRON-Netherlands Institute for Space Research algorithm (Hasekamp et al., 2011,
Fu et al., 2020) and generated by the GRASP (Generalized Retrieval of Atmosphere and Surface Properties) algorithm (Dubovik, 2014) make full use of multi-angle, multi-spectral polarimetric data. For example, Chen et al. (2018, 2019) use POLDER/GRASP spectral AOD and Aerosol Absorption Optical Depth (AAOD) to estimate emissions of desert dust, BC, and OC. Similarly, Tsikerdekis et al., (2021) use POLDER/SRON AOD, AAOD, EAE and SSA, but with a different model and assimilation technique,
and to estimate the aerosol mass and number mixing ratio of specific aerosol species.

On the other hand, AMTs inferred by various techniques and using satellite remote sensing observations are useful to provide spatial context (e.g., regional, seasonal, annual trends) to support other observations of aerosols and clouds or evaluate other aerosol type classifications. These AMTs are also useful in evaluating models in simple cases where a single aerosol species is present (e.g., "pure dust"). For
example, Johnson et al. (2012) demonstrated how CALIPSO mineral dust aerosol extinction retrievals were applied to improve dust emission and size distribution parameterizations in the global GEOS-Chem model, a global 3-D model of atmospheric chemistry driven by meteorological input from the Goddard Earth Observing System (GEOS).

We have inferred qualitative AMTs from passive POLDER/SRON remote-sensing retrievals of EAE
between 491 and 863 nm, SSA at 491 nm, a difference in Single Scattering Albedo, dSSA between 863nm and 491 nm, a Real Refractive Index, RRI at 670 nm and a pre-Specified Clustering and Mahalanobis Classification method (SCMC) (Russell et al., 2014).

The SCMC method, based on the methodology developed by Burton et al. (2012), uses the Mahalanobis distance (Mahalanobis, 1936) analysis in multidimensional space to assign AMTs based on a suite of
observed parameters. The number of parameters is adjustable, as are the nature of the parameters themselves. Similarly, the AMT definitions are flexible. However, a key requirement for the SCMC method is that reference values for each AMT must be defined (i.e., the mean, variances and covariances of the aerosol variables), typically using prescribed AMTs for a subset of observations. In practice, when applying SCMC to a new environment, a training data set is created by prescribing a set of air masses
based on independent observations. Those pre-specified AMTs from Russell et al. (2014) are based on dominant aerosol types from AErosol RObotic NETwork (AERONET) stations at specific locations and times (Holben et al., 1998). In Russell et al. (2014), qualitative AMTs were derived over the island of Crete, Greece, during a 5-year period using the SCMC method and pre-specified AMTs from global AERONET observations.


In this paper, we have extended the methods of Russell et al. (2014) (i.e., over Greece) to the entire globe for the year 2006. On the one hand, the POLDER-derived AMTs presented reassuring features such as (i) dust over the Atlantic between the Saharan coast and Central to South America, predominant in MAM and JJA, (ii) urban industrial aerosols found near industrialized cities such as the East Coast of North
America and over South East Asia, and (iii) two different types of Biomass Burning (BB) over the South East Atlantic (i.e., one illustrating more smoldering combustion and pre-specified using AERONET


stations located in South America and the other one illustrating more flaming combustion and using AERONET stations in Africa). We found darker BB (i.e., lower SSA) in August compared to September, due to an increase of POLDER-retrieved SSA during the season, reflecting either a change in BB aerosol
composition (Eck et al., 2013) or a mix of AMTs (Bond et al., 2013).

On the other hand, many features such as marine aerosols over the Saharan Desert or urban industrial aerosol type in South America, were most likely misclassified. Ambiguities in POLDER-derived AMTs could result from a combination of four factors:

(i)       Errors in POLDER reflectance/polarization measurements and aerosol retrievals (e.g., errors in
POLDER retrievals get larger for smaller AOD and/ or smaller range of scattering angles),

(ii)      A coarse spatial resolution of the gridded POLDER product (e.g., 2º x 2º),

(iii)      Non-optimal AERONET-based pre-specified AMTs used as a training dataset (e.g., the AMT illustrating more flaming combustion is defined in locations, such as Mongu in Africa, where smoldering and flaming combustion might be occurring at the same time, together with other AMTs present in the
atmospheric column) and/or

(iv)      A restricted number of POLDER-derived aerosol optical parameters. That is, the relative AMT discriminatory power increases with the number and diversity of observed parameters.

Unlike in Russell et al. (2014), where we used total column remote sensing-inferred optical properties which are often representative of a mix of different AMTs, the AMTs in this study are defined,
characterized, and derived using simultaneous gas-phase, chemical and optical instruments on the same aircraft. This reduces errors in measurements/retrievals and errors due to spatio-temporal colocation (i.e., i-ii). It also reduces ambiguities in the selection of the AMT training dataset (i.e., iii), and we specifically investigate the strengths and weaknesses of optical properties used as tools to define AMTs and how much these optical properties can capture dominant aerosol speciation (i.e., iv).


The objectives of this study are to:

•       Prescribe well-informed AMTs that display distinct aerosol chemical and optical signatures to act as a training (i.e., reference) AMT dataset, and

•       Evaluate the ability of airborne in-situ measured aerosol optical properties that are suitable to be
retrieved from space to successfully extract these AMTs.

## 2. Data and Method

### 2.1 Method

We select NASA DC-8 airborne *in-situ* data collected during the Study of Emissions and Atmospheric
Composition, Clouds, and Climate Coupling by Regional Surveys (SEAC$^4$RS) project (Toon et al., 2016), which was carried out in August–September 2013 over North America with a strong focus on the Southeastern US (SEUS). Measurements are collected at the altitude of the aircraft and are not representative of the full column satellite retrieval. Although these airborne *in situ* observations lack the



widespread coverage of surface networks or satellite retrievals, their benefits include measuring a wide
variety of gas-phase species, aerosol types and aerosol optical properties (Toon et al., 2016).

Figure 1 illustrates the overall method in this study. We proceed as follows:
1. We use gas phase and aerosol chemical (not optical) measurements to prescribe source-based AMTs
(called PS-AMTs) These measurements better characterize the aerosol properties in these AMTs
compared to observations of aerosol optical properties,
2. We investigate a suite of measured aerosol optical parameters for each PS-AMT, and then determine
the most useful and well separated aerosol optical properties,
3. We use the set of aerosol optical parameters defined in the second step above to define optical-based
class definitions (called DO-Class), including means, variances and covariances. In other terms, in this
step, we form the mathematical definitions of the classes,
4. We derive Optical-based AMTs (called DO-AMTs) using the set of aerosol optical properties defined
in the second step above, the DO-Class defined in the third step above and the SCMC method for a set of
observations that was not included in the training data sets,
5. We evaluate the ability of airborne aerosol optical properties to successfully extract PS-AMTs by
comparing PS-AMTs and DO-AMTs.
In Fig. 1, we illustrate AMTs as wolves, and their optical properties as their tracks. The second and
third step consist in describing the optical properties (or tracks) of each AMT (or wolf). The fourth step
consists in inferring an AMT (or wolf) from its optical properties (or tracks). The fifth and last step
consists in comparing the inferred to the initial AMT (or wolf).

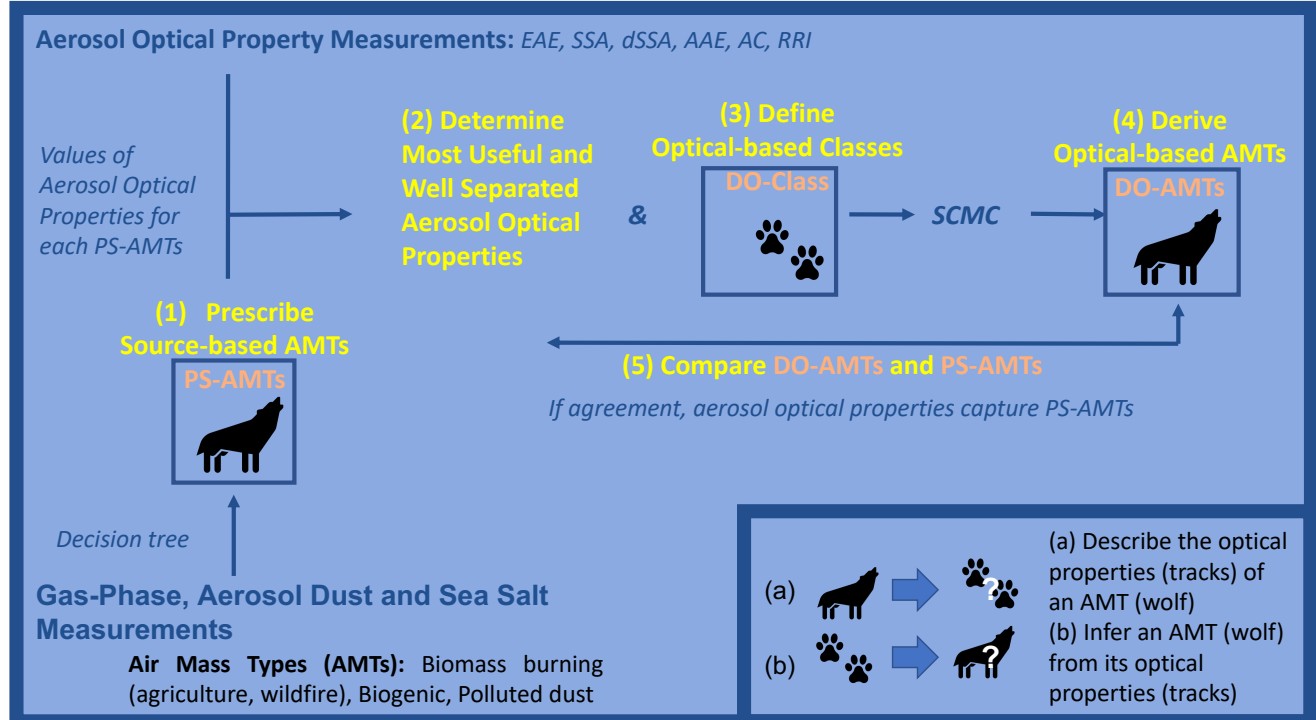

**Figure 1: Overall method in this study; PS-AMTs, DO-Class and DO-AMTs are Prescribed Source-based Air Mass Types (AMTs), Defined Optical-based Class definitions and Derived Optical-based AMTs. EAE: Extinction Angstrom Exponent; SSA: Single Scattering Albedo; dSSA: difference in**
**SSA; AAE: Absorption Angstrom Exponent; AC: absorption coefficient; RRI: real Refractive Index; SCMC: pre-Specified Clustering and Mahalanobis Classification. The concept of the wolf and its tracks is based on the dragon and its tracks in Bohren and Huffman (1983).**

In the first step of Fig.1, the PS-AMTs are defined using mostly gas measurements and a method based
on Espinosa et al. (2018) and Shingler et al. (2016) (see Fig. 2 and section A.1.3 in the appendix for more information). Section 3.1 describes these PS-AMTs, their location and composition in further detail during SEAC[4]RS.





**Figure 2: Classification scheme for pre-specifying Air Mass Types (PS-AMTs) using mostly gas**
**measurements and a method based on Espinosa et al. (2018) and Shingler et al. (2016) and modified**
**to include Marine and two different types of BB AMTs. See section A.1.3 in the appendix for more**
**information.**

In the second step of Fig.1, once the PS-AMTs are defined, we test whether these PS-AMTs exhibit
distinct aerosol optical properties and then, select the most useful and well separated aerosol optical
properties. To select the most useful and well separated aerosol optical properties for each PS-AMT, we
define a cluster in multi-dimensional parameter space, which is composed of all the data points (values
of optical properties) in that PS-AMT category. Then, for each point in the data set, we calculate the
nearest cluster using the Mahalanobis distance (Mahalanobis, 1936). If the nearest cluster to a point
corresponds to the PS-AMT, then that point is "steady". Let us note that this method was used in previous
studies (e.g., Espinosa et al. (2018)) and is described in further detail in section A.1.4 in the appendix.
Section 3.2 describes the results from this step i.e., the most useful and well separated aerosol optical
properties in our study.

In the third step of Fig.1, the DO-Class use the first "steady" (i.e., well separated) half of all valid aerosol
optical observations. Once the training (or reference) clusters DO-Class are defined, we use the
Mahalanobis distance to filter outliers from our training dataset and further "purify" them. Similar to
Russell et al. (2014), we delete points that have less than 1% probability of belonging to each pre-specified
DO-Class. We also delete from a specified cluster any points that are closer (in terms of Mahalanobis
distance) to a different cluster. Note that unlike in Russel et al. (2014), this additional filtering step (to the
"steady" filtering step) has a minimal impact on the training dataset in our study.



In the fourth step of Fig.1, a test data set is analyzed and classified. This test data set is based on independent observations and must be of the same nature than the training dataset. In this study, our test dataset is composed of independent airborne in-situ optical properties. It is the other half of all valid
aerosol optical observations (DO-Class are defined using the "steady" portion of the first half). We derive optical-based AMTs (DO-AMTs) to each test data point using the SCMC method and the DO-Class. This is achieved by assigning the test datapoint to the DO-Class that shows minimum Mahalanobis distance in a multi-dimensional space made of the best suited and most separable optical properties. We refer the reader to section 2 of Russell et al. (2014) or Burton et al. (2012) for a thorough description of the SCMC
method. Section 3.3 describes the results from these steps i.e., the defined optical- and derived optical-based air mass types in our study.

In the fifth and last step of Fig. 1, we evaluate the ability of airborne aerosol optical properties to successfully extract PS-AMTs by comparing the PS-AMTs and DO-AMTs. Section 3.4. describes the results of this final step in our study.

## 2.2 Instruments and Observations

A major strength of our study is the use of *in situ* gas-phase, chemical and optical instruments on the same NASA DC-8 research aircraft during the SEAC[4]RS campaign. Table 1 lists the various airborne *in situ* instruments and products used in this study. It also shows the size of the aerosol sampled by each
instrument, the way we use the products in our study (i.e., step 1 through 4 in Fig. 1) and important references for each instrument.

Note that instead of simply using the standardized SEAC[4]RS merged dataset, a lot of effort was dedicated to carefully collocate, combine (section A.1.2 in the appendix), cloud-screen, filter, humidify (i.e., converted from dry to ambient conditions), compute and interpolate/extrapolate optical parameters to
specific wavelengths (section A.1.1 in the appendix).



| | Instruments | Products | Sampled Aerosol Size | Usage | References |
|---|---|---|---|---|---|
| 1 | PTR-MS, DACOM, TD-LIF, NO$_y$O$_3$ | Acetonitrile, isoprene, monoterpene, CO, NO$_2$ | - | (1) | PTR-MS (Mikoviny et al., 2010); DACOM (Fried et al., 2008); TD-LIF (Cleary et al., 2002); NO$_y$O$_3$ (Ryerson et al., 2012) |
| 2 | PALMS | Internally mixed Sulfate/ Organic/ Nitrate (SON), Biomass Burning (BB), Sea salt, and Dust particle types | <5µm dry diameter | (1), (*) | Murphy et al., 2006 Froyd et al., 2019 |
| 3 | SAGA | Cl, Br, NO$_3$, SO$_4$, C$_2$O$_4$, Na, NH$_4$, K, Mg, Ca | <4µm dry diameter | (*) | Dibb et al., 2003 |
| 4 | AMS | OA, sulfate, ammonium, nitrate | 0.02 - 0.8 µm (trapezoidal transmission efficiency, D50 at 0.035 and 0.35 µm) | (*) | DeCarlo et al., 2006; Canagaratna et al., 2007; Hu et al., 2015; Guo et al., 2021 |
| 5 | SP2 | BC | 0.1-0.5µm (BC component, only) | (*) | Perring et al., 2017 |
| 6 | LARGE TSI and PSAP | Absorption, Scattering and Extinction Coefficient (AC, SC and EC) at 450, 550 and 700 nm | <5µm dry diameter for Dry Total Scattering Coefficients at 450, 550, and 700 nm (TSI Neph) and Total Absorption Coefficients at 467, 530 and 660 nm (PSAP) | (2-3-4) | Ziemba et al., 2013; McNaughton et al., 2007 |
| 7 | DASH-SP | Real Refractive Index (RRI) at 532nm | 0.18-0.40µm dry diameter | (2-3-4) | Sorooshian et al., 2008; Shingler et al., 2016 |
| 8 | PI-Neph | Real Refractive Index (RRI) at 532 nm | <5 µm dry diameter | (2-3-4) | Dolgos and Martins, 2014; Espinosa, 2017, 2018 |

Usage (see step 1-4 in Fig. 1):
    (1)  Pre-specify PS-AMTs
    (*)  Verify/ further define PS-AMTs
    (2-3-4) Derive DO-AMTs to assess ability of aerosol optical properties to observe PS-AMTs

**Table 1: Instruments, products, sampled aerosol size, usage and references relevant to this study. More information on the instruments during SEAC$^4$RS can be found here: https://espo.nasa.gov/home/seac4rs/content/Instruments.**

The first step in Fig. 1 (i.e., prescribe source-based PS-AMTs) uses the gas-phase and aerosol instruments in line 1-2 of Table 1. The following steps in Fig. 1 (i.e., define the most useful and well separated optical





properties, define optical-based classes and derive optical-based AMTs) use the optical instruments in lines 6-8 of Table 1. Let us emphasize that the instruments in Table 1 sample different aerosol sizes. This is especially true for the DASH-SP instrument, which sampled particles with dry diameters between 180 and 400 nm during SEAC$^4$RS (Shingler et al., 2016). In contrast, the sampled air was provided to the PI-Neph instrument through the NASA LARGE shrouded diffuser inlet, which sampled isokinetically and is known to have a 50% passing efficiency at an aerodynamic diameter of at least 5 μm at low altitude (McNaughton et al., 2007; Espinosa et al., 2017).

In this study, we use the sixteen aerosol optical parameters listed in Table 2 (i.e., six parameters at 3 wavelengths and/or 3 combinations of wavelengths) and derived from the optical instruments in line 6-8 of Table 1. These optical parameters were computed from the initial measurements and using the equations listed in the second column of Table 2 (see section A.1.1 in the appendix for more information on these calculations).

| Aerosol Optical Parameters | Computed Using |
|---|---|
| Extinction Angstrom Exponent, EAE | LARGE EC and Eq. 7.1.1.a |
| Absorption Angstrom Exponent, AAE | LARGE AC and Eq. 7.1.1.c |
| Single Scattering Albedo, SSA | LARGE EC, SC and Eq. 7.1.1.b |
| Difference in SSA at two wavelengths, dSSA | LARGE EC, SC and Eq. 7.1.1.b |
| Absorption Coefficient, AC | LARGE EC, SC and Eq. 7.1.1.d |
| Real Refractive Index, RRI | PI-Neph or DASH-SP and LARGE |

**Table 2: EC, SC and AC stand for Extinction, Scattering and Absorption Coefficients i.e., *in situ* aerosol optical parameters provided at a given aircraft altitude in this study. Wavelengths are (i) 450, 550 and 700 nm for SSA and AC and (ii) 450-550, 550-700 and 450-700 nm for EAE, AAE and dSSA.**

## 3. Results

### 3.1 Prescribe Source-based Air Mass Types (PS-AMTs)

Figure 3 shows the PS-AMTs pre-specified using mostly measured gas phase compounds and the method described in Fig. 2.



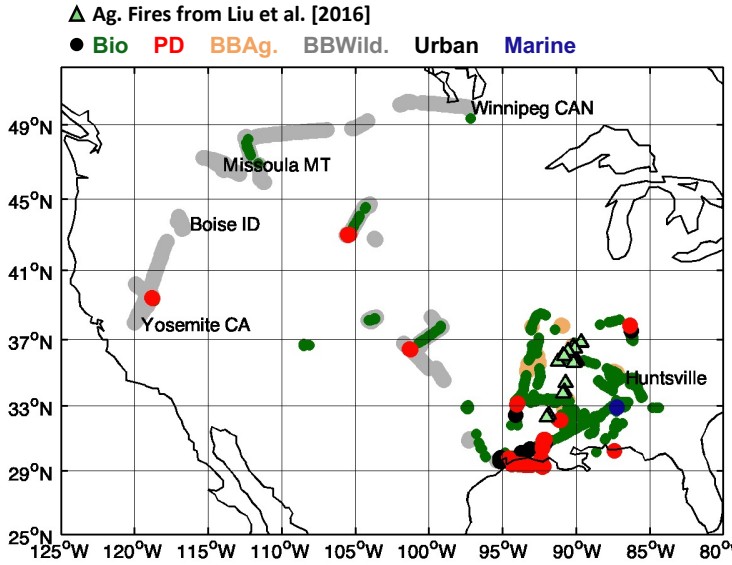

**Figure 3: Air mass types pre-specified (PS-AMT) using mostly gas measurements and methods based on Espinosa et al. (2018) and Shingler et al., (2016) (see Fig. 2). The number of data points assigned to each PS-AMTs are N=31 BBAg., N=382 BBWild., N=646 Bio. and N=46 PollDust PS-AMTs. PS-AMTs Marine and Urban were not analyzed in the remainder of this study due to their limited number of data points (Urban in black shows N=9 and Marine in blue shows N=7). Green triangles show the location of agricultural fires according to Liu et al. (2016).**

During SEAC$^4$RS, according to Kim et al. (2015) and Wagner et al. (2015), the campaign-averaged aerosol mass was composed of mostly Organic Aerosol (OA) that is internally mixed with sulfate and nitrate at all altitudes over the southeastern U.S (SEUS) i.e., 55% OA and 25% sulfate mass on average according to ground-based filter-based PM$_{2.5}$ (Particulate Matter concentration with an aerodynamic diameter smaller than 2.5 μm) speciation measurements from EPA CSN sites. This is consistent with the

findings of Edgerton et al. (2006), Hu et al. (2015), Xu et al. (2015) and Weber et al. (2007) which show that PM2.5 is dominated by SOA and sulfate during the summer in SEUS. Aircraft data show that 60% of the aerosol column mass (i.e., mostly OA and sulfate) is contained within the mixing layer (Kim et al., 2015).

GEOS-Chem attributes OA mass as 60% from biogenic isoprene and monoterpenes sources (with a

significant role of isoprene in accordance with Hu et al. (2015), Marais et al. (2016), Zhang et al. (2018), Jo et al. (2019), and Liao et al. (2015)), 30% from anthropogenic sources and 10% from open fires (Kim et al., 2015). Espinosa et al. (2018) confirms the domination of biogenic emissions in the SEUS (see their Fig. 2). Fig. 3, in agreement with these studies, shows a majority of biogenic PS-AMTs (in green, N=646), mostly in the SEUS during SEAC$^4$RS.

During SEAC$^4$RS, the air sampled by the DC8 was also affected by both long-range transport of wildfire from the west (Peterson et al., 2015; Saide et al., 2015; Forrister et al., 2015; Liu et al., 2017) and local agricultural fires mostly from the burning of rice straw along the Mississippi River Valley (Liu et al., 2016). Fig. 3, in agreement with these studies, shows BBWild PS-AMT in the West (in grey, N=382) and BBAg PS-AMT in the East (in salmon, N=31). Both agricultural and wildfire smoke is mainly composed

of OA, which includes a substantial amount of light-absorbing brown carbon, BrC (Liu et al., 2017), produced mostly by smoldering combustion (Reid et al., 2005; Laskin et al., 2015).





Although Fig. 3 also shows Urban and Marine PS-AMTs in the SEUS, these PS-AMTs were not further analyzed in the remainder of this study due to their limited number of data points (Urban in black with N=9 and Marine in blue with N=7 data points).

Figure 4 describes the aerosol chemical signatures of the principal PS-AMTs using the PALMS, SAGA, AMS and SP2 instruments (see line 2-5 in Table 1 for more information on these instruments and their products). Note that some aerosol components (e.g., Organic, Sulfate, Nitrate) are very general chemical indicators and much less specific than the gas-phase chemistry they are trying to predict. These aerosol components are nonetheless directly comparable to aerosol chemical components simulated in chemical

transport (CTM) (e.g., GEOS-Chem, the Goddard Chemistry, Aerosol, Radiation, and Transport model, GOCART, the Weather Research and Forecasting model coupled with Chemistry, WRF-Chem) and air quality (AQ) models (e.g., the Community Multiscale Air Quality Modeling System, CMAQ).




**Figure 4: (a) Average PALMS normalized volume concentration per PS-AMT. PALMS normalization uses the sum of BB particles, sulfate-, organic- and nitrate-rich particles from non-BB sources, mineral dust, sulfate-organic-nitrate (SON) particles without a dominant sub-**



**component, and sea-salt (the latter two PALMS aerosol types are not shown and constitute the remainder). (b) Averaged and normalized SAGA mass concentrations per PS-AMT; normalization uses the sum of all the SAGA components in the x-axis (c) Normalized mass fractions of AMS sulfate, ammonium, nitrate, OA, SP2 BC and ratio of SP2 BC and AMS OA per PS-AMT. The AMS inorganic mass fraction of sulfate, ammonium and nitrate are normalized to the sum of**
**sulfate, ammonium, and nitrate. The AMS and SP2 total Non-Refractory NR-mass fraction of OA and BC are normalized to the sum of OA, BC, sulfate, ammonium, and nitrate. In each blue box, the red horizontal line indicates the median, and the bottom and top edges of the box indicate the 25th and 75th percentiles, respectively. The black whiskers extend to the most extreme data points not considered outliers, and the outliers are plotted individually using red points. PS-AMTs Marine**
**and Urban are not analyzed due to their limited number of data points (N=9 Urban and N=7 Marine PS-AMTs).**

Note that the four aerosol instruments in Fig. 4 measure different aerosol properties. For instance, AMS and SAGA measure bulk concentrations of chemical sub-components (e.g., sulfate) whereas
PALMS classifies individual particles into several size-resolved types, including mineral dust, BB and several non-BB types that have varying amounts of internally mixed sulfate, organic, and nitrate.

The PS-AMTs on Fig. 4 show expected chemical features:
• The BB PS-AMTs (i.e., BBAg. and BBWild.) record high BB particle concentrations from
PALMS in Fig. 4a, high nitrate (Nit), ammonium (Amm), calcium (Ca) and potassium (K) concentrations from SAGA in Fig. 4b, high OA (i.e., >0.8) from AMS and high BC mass fractions from SP2 in Fig. 4c, in agreement with many other studies (e.g., Cubison et al., (2011); Hecobian et al., (2011); Jolleys et al., (2015), Guo et al. (2020)). The BB PS-AMTs also record higher AMS ammonium and nitrate, compared to Bio. and PollDust PS-AMTs in Fig. 4c. This is due to ammonium nitrate (NH4NO3) forming in fires
by neutralization of freshly formed nitric acid from NOx oxidation with an excess of primary ammonia (e.g., Guo et al. (2020)).
• The Bio. PS-AMTs record higher non-BB organic-rich particles from PALMS in Fig. 4a, higher SAGA sulfate concentrations in Fig. 4b, smaller nitrate and ammonium (i.e., relatively acidic) and higher sulfate particle concentrations (from e.g., coal plants) from AMS in Fig. 4c, compared to the BB PS-
AMTs. As such, the Bio. PS-AMTs in this study are typical of the SEUS region (e.g., (Kim et al., 2015 and Hu 2015)). When using Positive Matrix Factorization (PMF) (Ulbrich et al., 2009) on the AMS measurements, most of the organic aerosols in the Bio. PS-AMTs is composed of biogenic SOA. The Bio. PS-AMTs also record significantly lower BC concentrations from the SP2 as well as BC to OA ratios from the AMS and SP2 in Fig. 4c, compared to the BB and PollDust PS-AMTs, in accordance with e.g.,
Hodzic et al. (2020).
• The PollDust PS-AMTs record, as expected, high dust concentration from PALMS in Fig. 4a and high calcium (Ca) and magnesium (Mg) from SAGA in Fig. 4b. In addition, the PollDust PS-AMTs also include BB from PALMS in Fig. 4a and possibly a minor sea salt component (i.e., high Na and Cl) from SAGA in Fig. 4b as well as relatively high sulfate from SAGA and AMS in Fig. 4c. A compositional
picture of the PollDust PS-AMTs from PALMS in section A.2.3 in the appendix shows dust



predominately in the coarse mode but also an accumulation mode that contains a variety of particle types, all of which contain sulfate and organic material.

The analysis in Fig. 4 confirms that the gas-phase-derived PS-AMTs indeed have distinct aerosol chemical properties. Therefore, we explore whether these PS-AMTs can be derived using only aerosol optical properties.

## 3.2 Determine Most Useful and Well Separated Aerosol Optical Properties

As described in section 2.1, we need to test if the PS-AMTs (from section 3.1) exhibit distinct aerosol optical properties. This is an essential first step to optimize the final prediction of AMTs using aerosol optical properties (DO-AMTs).

We start with the sixteen aerosol optical parameters in Table 2 (i.e., EAE, SSA, dSSA, AAE and AC at different combinations of 450, 550 and 700 nm and RRI at 532 nm). Section A.2.1 in the appendix
illustrates the ranges of these sixteen aerosol optical parameters, classified by PS-AMTs. Given that many of these parameters have similar properties, we select six out of these sixteen aerosol optical parameters, to simplify the analysis and presentation of results. To do that, we first look at the percentage of points unambiguously retrieved or "steady" (i.e., points that are well separated from other clusters and, hence, remain in their initial clusters) when using different combinations of two out of sixteen aerosol optical
parameters across all four PS-AMTs. We first select parameters AAE between 450 and 550nm and RRI at 532nm as they form the only combination of two parameters to achieve >65% "steady points" for all four PS-AMTs (see Fig. A5 in the appendix). The rest of the six optical parameters are either chosen at 550nm (i.e., closest wavelength to 532 nm) or between 450 and 550nm. As a result, the six parameters we choose for the remainder of this study are dSSA 450-550 nm, RRI 532 nm, EAE 450-550 nm, AAE
450-550 nm, SSA 550 nm and AC at 550nm. Among these parameters, the usefulness of parameters dSSA 450-550 nm, EAE 450-550 nm, SSA 550 nm and AC at 550nm only becomes apparent in a 3-D parameter space (see Fig. A6 and its orange boxes in the appendix, which record >65% "steady points" for many combinations of three parameters among these six selected aerosol optical parameters).

Figure 5 illustrates the range of these six aerosol optical properties for each PS-AMT. Fine particles (i.e., BBWild., BBAg. and Bio. PS-AMTs with higher EAE values) show mostly well-separated variability in RRI, AAE and dSSA. Coarse particles (i.e., PollDust PS-AMT with lower EAE values) is optically distinctive from the other PS-AMTs, particularly showing lower RRI, higher AAE and higher dSSA. In agreement with Selimovic et al. (2019; 2020) in Missoula, MT, we seem to also observe separate optical
signatures, and more specifically different AAE ranges, for BBAg. and BBWild. PS-AMTs during SEAC⁴RS.
The aerosol optical properties of the PollDust PS-AMTs in this study differ from the ones of the "pure dust" AMT in Russel et al. (2014). The "pure dust" in Russel et al. (2014) is based on AERONET measurements in various dusty regions of the world. In this study, PollDust PS-AMT show a median EAE
of ~1.3 between 450 and 550 nm and a median RRI of ~1.4 at 532 nm on Fig. 5, compared to respectively





~0 between 491 and 864 nm and 1.53 at 670 nm for AERONET-based "pure dust" in Russel et al. (2014). We show that the higher PollDust PS-AMT EAE values in our study are due to the presence of accumulation mode non-dust aerosols, which constitute a significant contribution to the total number and volume concentration of particles (see section A.2.3 in the appendix for a compositional picture of PollDust PS-AMT). Similarly, we also suggest that the low PollDust PS-AMT RRI values are due to its non-dust accumulation mode, which is generally more hygroscopic than pure dust and may have a larger contribution to the PollDust total Growth Factor (GF, see Eq. 7.1.1e). We refer the reader to section A.2.1 in the appendix for a closer look at RRI values in the case of PollDust PS-AMTs from the PI-Neph and DASH-SP instruments separately.

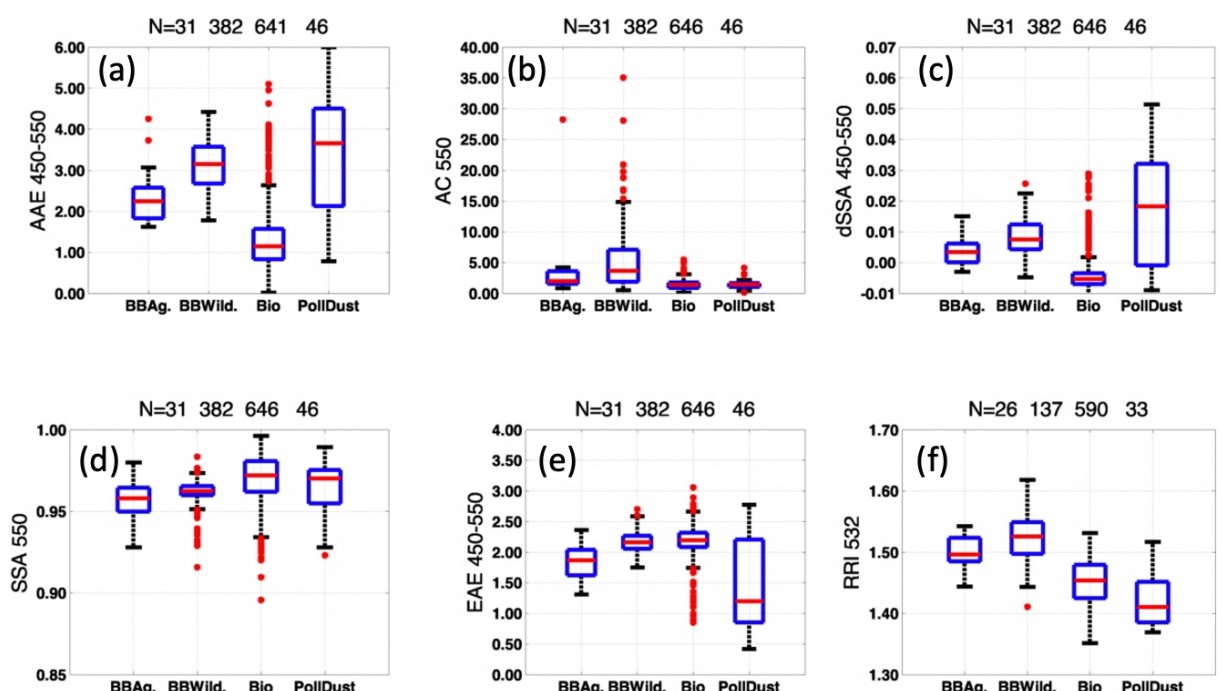

**Figure 5: Optical characterization of PS-AMTs using the LARGE, PI-Neph and DASH-SP instruments (see Table 1). In each blue box, the red horizontal line indicates the median, and the bottom and top edges of the box indicate the 25th and 75th percentiles, respectively. The black whiskers extend to the most extreme data points not considered outliers, and the outliers are plotted individually using red points. AAE: Absorption Angstrom Exponent, AC: Absorption Coefficient, dSSA: difference in Single Scattering Albedo, SSA: Single Scattering Albedo, EAE: Extinction Angstrom Exponent, RRI: Real Refractive Index. Numbers in the title correspond to the number of points behind each box-whisker for the respective BBAg., BBWild., Bio. and PollDust PS-AMTs.**

Figure 6 shows "steady" values (i.e., fraction of cases of a given type that are correctly identified; see section 2.1) for combinations of two, three and four optical parameters out of the six selected aerosol





optical parameters (see Fig. 5) and four AMTs (i.e., BBAg., BBWild., Bio. and PollDust). Moving
forward, we select the sixteen combinations of optical parameters highlighted by grey boxes and black
dots in Fig. 6, as they show > 65% "steady points" (i.e., successfully separate aerosol signatures) for PS-
AMTs BBAg., BBWild., Bio. and PollDust. These combinations are shown as black squares in the table
of Fig 8.


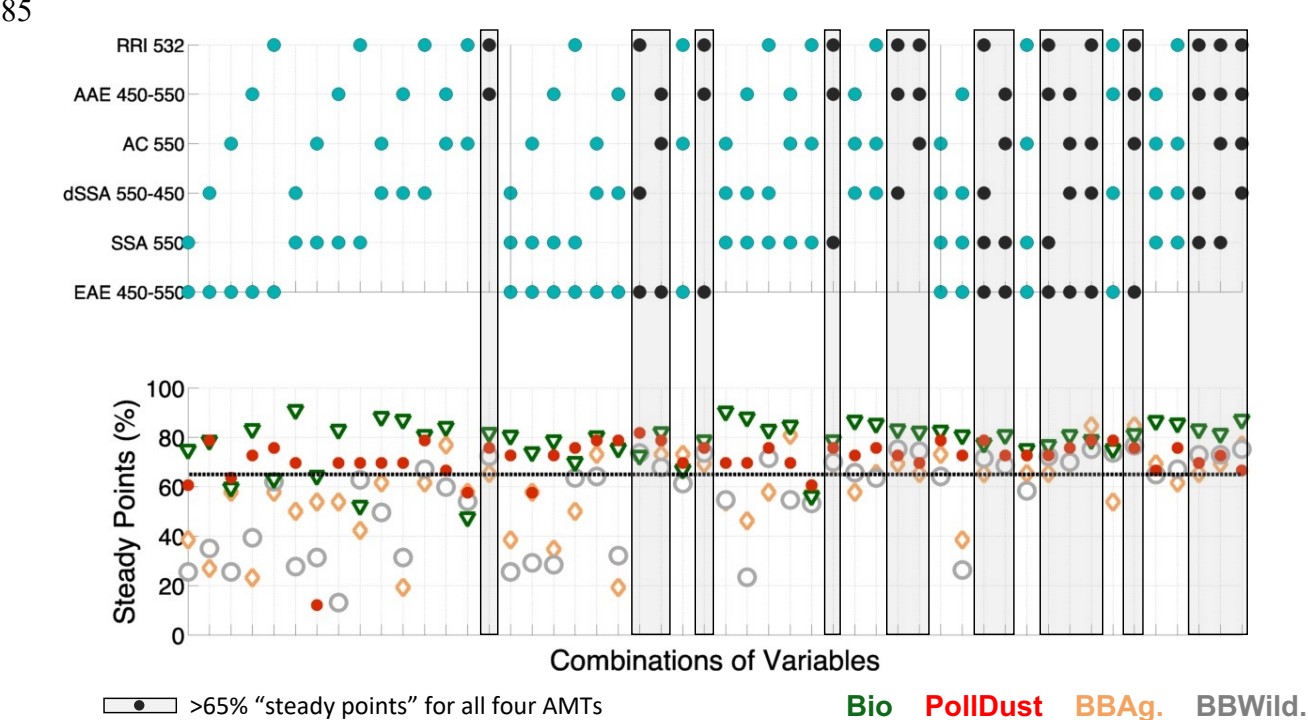

**Figure 6: Percentage of points "steady" (i.e., fraction of cases of a given type that are correctly
identified; see section 2.1) in the lower panel when using different combinations of aerosol optical
parameters in the upper panel for each PS-AMT. Grey boxes and black points depict combinations
of optical parameters showing > 65% "steady points" for PS-AMTs BBAg., BBWild., Bio. and
PollDust. RRI: Real Refractive Index, AAE: Absorption Angstrom Exponent, AC: Absorption
Coefficient, dSSA: difference in Single Scattering Albedo, SSA: Single Scattering Albedo, EAE:
Extinction Angstrom Exponent**

Let us note that for some cases, the fraction of "steady" points seems to decrease when adding classifying
variables. These cases were investigated and are mostly due to fewer data points that are non- "steady"
when adding classifying parameters, out of an already small total number of datapoints (e.g., a
combination of EAE, dSSA, AAE and RRI show <65% "steady points" for BBAg. PS-AMT, compared
to >65% "steady points" for a combination of EAE, AAE and RRI; this is due to 4 more "steady" points





(N=18) when using a combination of 3 parameters, compared to 4 parameters (N=14), out of a total of N=26 cases).

Moreover, we suggest that higher aerosol loadings within the air masses allow for more accurate identification by optical properties, due to higher accuracy of the aerosol optical properties themselves.
For example, we have seen an increase from ~80% to 100% "steady" data points in the BBWild. PS-AMT when using EAE, AAE and RRI when extinction coefficients increased from 30-40 m$^{-1}$ Mm$^{-1}$ to 60-70 Mm$^{-1}$ (number of data points between N=11 and N=20).

**3.3 Define Optical-based Class Definitions and Derive Optical-based Air Mass Types (DO-Class**
**and DO-AMTs)**

Our goal in this section is to derive AMTs (DO-AMTs), followed by a comparison between DO-AMTs and the initial PS-AMTs to test the ability of aerosol optical properties alone to capture PS-AMTs.
As described in section 2.1, to derive DO-AMTs using the SCMC method, we need (i) a combination of
useful and well separated optical properties (e.g., EAE, AAE and RRI or combination #4 in Table of Fig. 8), (ii) a set of defined classes or clusters of reference (i.e., a training dataset that we call DO-Class) and (iii) the computation of the Mahalanobis distance between each observation we want to classify in a test data set and each of the clusters from the training dataset.

We introduce Table 3, which records the number of data points behind each step in our study.



| Number of Data | BBAg | BBWild | Bio. | PollDust | Total | Major Steps (see Fig. 1) |
|---|---|---|---|---|---|---|
| 1 PS-AMTs | 31 | 382 | 646 | 46 | 1105 | **(1)** Pre-specify Source-based PS-AMTs (see colored points in Fig. 3) |
| 2 Valid AAE (Fig. 5a) | 31 | 382 | 641 | 46 | 1100 | **(2)** Determine most useful and well separated aerosol optical properties |
| 3 Valid RRI (Fig. 5f) | 26 | 137 | 590 | 33 | 786 | |
| 4 Valid EAE, AAE and RRI Observations (#4 in Fig. 8) | 26 | 137 | 585 | 33 | 781 | |
| 5 "Steady" points (*) | 18 | 101 | 460 | 25 | 604 | |
| 6 To define DO-Class | | | 391 | | | **(3)** Define optical-based Classes, DO-Class; use "steady" portion of first ~half of observations |
| 7 DO-Class (*) | 8 | 52 | 238 | 13 | 311 | |
| 8 To derive DO-AMTs | | | 389 | | | **(4)** Derive optical-based AMTs. DO-AMTs; apply SCMC method, using DO-Class, on second ~half of observations |
| 9 Known DO-AMTs (*) | 32 | 55 | 217 | 77 | 381 | |
| 10 Unknown DO-AMTs (*) | | | 8 | | | |
| 11 PS-AMTs | 13 | 68 | 292 | 16 | 389 | **(5)** Compare DO-AMTs and PS-AMTs |
| 12 DO-AMTs similar to PS-AMTs (*) | 10 | 54 | 213 | 13 | 290 | |
| 13 DO-AMTs similar to PS-AMTs as a % of assigned DO-AMTs (l9) (*) | 31 | 98 | 98 | 17 | - | |
| 14 DO-AMTs similar to PS-AMTs as a % of PS-AMTs (l11) (*) | 77 | 79 | 73 | 81 | - | |

(*) in the case of combination #4 in Fig. 8 i.e., EAE, AAE and RRI

**Table 3: Number of data points per AMTs behind each step in our study. PS-AMTs Marine and Urban are not analyzed due to their limited number of data points (N=9 Urban and N=7 Marine PS-AMTs). EAE: Extinction Angstrom Exponent, AAE: Aerosol Absorption Exponent and RRI: Real Refractive Index.**

The first line of Table 3 shows the number of data points per PS-AMTs (see section 3.1). Then, it shows the valid number of data points behind AAE (Fig. 5a), RRI (Fig. f) and a combination of EAE, AAE and RRI (see respectively line 2, 3 and 4 of Table 3). It also shows the "steady" number of data points per PS-AMT in line 5 in the case of a combination of EAE, AAE and RRI (see Fig. 6).

To create the training data set DO-Class (line 7 in Table 3), we select the "steady" portion of half (every other sample) of the entire set of valid datapoints (line 6 of Table 3). The test data set that we want to classify as DO-AMTs is the other half of the entire set of valid datapoints (line 8 in Table 3). This DO-AMT dataset is made of "steady" and non- "steady" data points.





Figure 7 illustrates the separability of the DO-Class in the 3-D space made of aerosol optical parameters EAE, AAE and RRI. The regions of the DO-Class are described by colored ellipses representing the mean,
variance, and covariance of the DO-Class training set. It also shows that most of the DO-Class represent the original source-based PS-AMTs (represented by colored triangles on Fig. 7). However, let us note that a distinct portion of the Bio. PS-AMTs (green triangles) seem to not be represented by the Bio. DO-Class (green ellipse). These Bio. PS-AMTs show higher AAE and lower EAE values and mostly fall into the PollDust DO-Class instead (red ellipse).


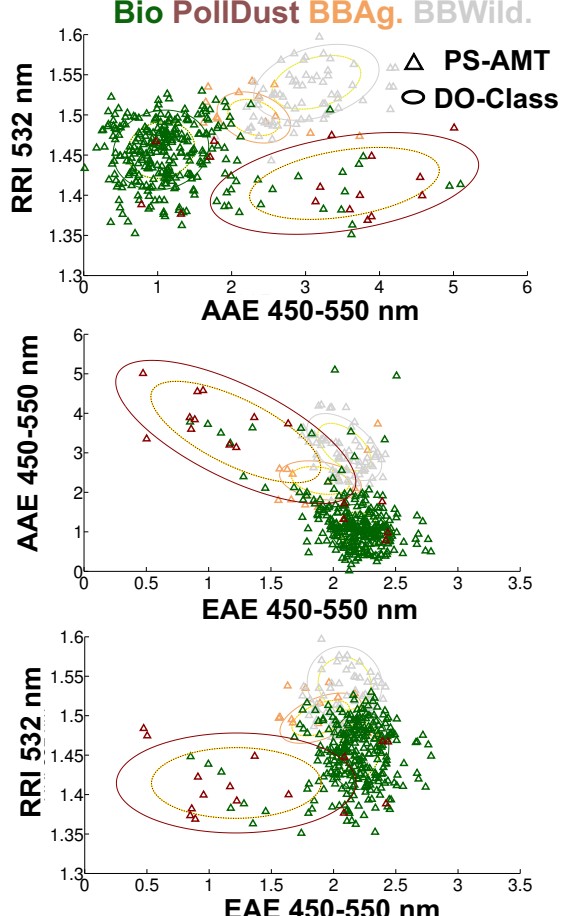

**Figure 7: DO-Class definition (solid and dashed ellipses colored by AMTs defining boundaries of the DO-Class clusters; no DO-Class data points are plotted) and prescribed source-based PS-AMTs (triangles colored by AMTs). 75% of the DO-Class are contained in the solid ellipses and 50% of the DO-Class are contained in the dashed ellipses. RRI: Real Refractive Index, AAE: Absorption Angstrom Exponent, EAE: Extinction Angstrom Exponent.**

Line 9 in Table 3 shows the number of DO-AMTs (correctly and incorrectly) classified as BBAg., BBWild., Bio. or PollDust AMTs using the combination of EAE, AAE and RRI as an example, the SCMC method and the DO-Class reference clusters. Most points from the test data set were assigned an AMT (see N=381 assigned DO-AMTs on line 9, compared to N=8 unknown on line 10 of Table 3). Unclassified/unknown DO-AMTs are those where the 3-D data point is outside the 99% probability surface for all four DO-Classes.

### 3.4 Compare Optical-based Compared to Source-based Air Mass Types (DO- vs. PS-AMTs)

Once we have derived DO-AMTs from optical properties (i.e., inferred our wolf based on its tracks in Fig. 1), we need to assess how many of the DO-AMTs agree with those originally assigned as PS-AMTs. Line 11 in Table 3 shows the number of prescribed PS-AMTs in each category when only looking at the test dataset to derive DO-AMTs on line 8 of Table 3 (N=389). Line 12 in Table 3 shows the number of DO-AMTs
that are identical to PS-AMTs. Line 13 and 14 show the same result, but as a percentage of the respectively derived DO-AMTs or prescribed PS-AMTs in the same category. In Table 3, we find 77% BBAg., 79%





BBWild., 73% Bio. and 81% PollDust PS-AMTs are correctly reflected in the DO-AMTs. This result can also be seen for combination #4 in Fig. 8 (i.e., EAE, AAE and RRI).

Fig. 8 illustrates the percentage of identical DO-AMTs to PS-AMTs when using each of the 16 combinations of optical parameters illustrated by black squares in the table of Fig. 8. This percentage, like line 14 in Table 3, is computed as the number of DO-AMTs that agree with those originally assigned as PS-AMTs, compared to the total number of prescribed PS-AMTs in each category in our test dataset (e.g., line 11 in Table 3).

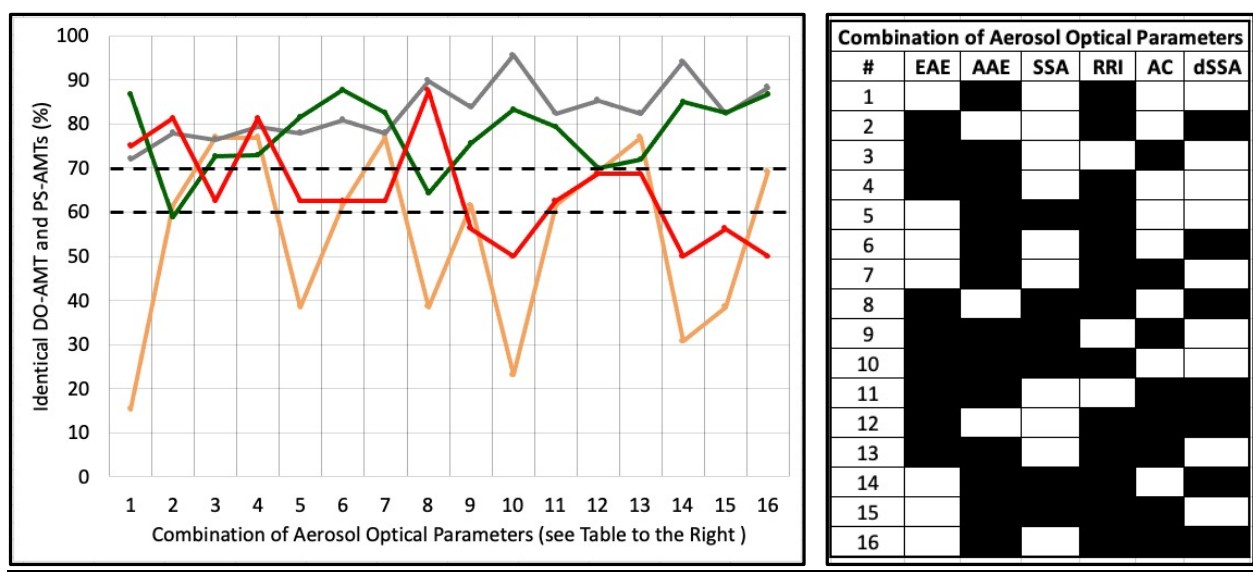


**Figure 8: Identical DO-AMT and PS-AMTs as a percentage of prescribed PS-AMTs in each category when using the different combinations of optical parameter listed in the table to the right (black squares show combination on each line) and for the four PS-AMTs BBAg. (salmon), BBWild. (grey), Bio. (green) and PollDust (red). Back horizontal dashed lines show 60% and 70% identical**
**DO-AMT and PS-AMTs.**

According to Fig. 8, the entire sixteen combinations of aerosol optical properties EAE, AAE, SSA, RRI, AC and dSSA listed in the Table of Fig. 8 as black squares seem to capture both the Bio. and BBWild. PS-AMTs (>~60% identical DO-AMT and PS-AMTs in green and grey solid lines in Fig. 8). We remind
the reader that these PS-AMTs are mostly based on gas measurements (see Fig. 2) and are dominated by different aerosol species (see Fig. 4).

On the other hand, fewer combinations of aerosol optical parameters seem to adequately capture the BBAg. and PollDust PS-AMTs. Further analysis shows that, in average, most DO-AMTs assigned to the BBAg. and PollDust categories are, in fact, misclassified and failing to capture the Bio. PS-AMTs. As
shown earlier in Fig. 7, we suggest these DO-AMTs fail to capture the Bio. PS-AMTs because the Bio. DO-Class might not be entirely representative of the Bio. PS-AMTs (see green triangles outside of the green ellipses in Fig. 7).

Note that three combinations of aerosol optical parameters, namely #4 (EAE, AAE and RRI), #12 (EAE, RRI AC and dSSA) and #13 (EAE, AAE, RRI and AC) in Fig. 8, seem to capture all four PS-AMTs



particularly well (>~70% identical DO-AMT and PS-AMTs). Let us mention that results linked to the use
of the absorption coefficient, AC, an extensive property that is dependent on aerosol loading, is likely to
be unique to this study and might not be representative of any other field campaign.

## 4. Conclusion

One desire of our scientific community is to ultimately translate the space-based "total atmospheric
column effective" AMTs such as biomass burning, dust, urban industrial, and polluted marine into
chemical species with defined emission source inventories and formation/aging chemistry such as sulfate,
BC, OA, SOA, nitrate, dust, or sea salt to better improve models. Fully achieving that goal might not be
feasible and progress can only be incremental. This study constitutes a first step (many steps remain)
towards the goal of translating the space-based "total atmospheric column effective" aerosol optical
properties and derived optical-based AMTs into source-based AMTs.

Current satellite derived AMTs inferred by various techniques are useful to provide spatial context to
support other observations of aerosols and clouds or evaluate other aerosol type classifications. However,
these satellite derived AMTs are subjectively/ambiguously defined and might often be misclassified.
The AMTs in this study are defined, characterized, and derived using gas-phase, chemical and optical
instruments on the same aircraft. This reduces errors in measurements/retrievals, due to spatio-temporal
colocation and ambiguities in the selection of the AMT training dataset. We also specifically investigate
the strengths and weaknesses of various aerosol optical properties used as tools to define AMTs and how
much these optical properties can capture dominant aerosol speciation.

We first define AMTs using mostly airborne gas-phase measurements during SEAC⁴RS. We find distinct
optical signatures for biomass burning (from agricultural/ prescribed or wildfires), biogenic and dust-
influence AMTs (Marine and Urban AMTs show too few data points to analyze). Useful aerosol optical
properties to characterize these signatures are the extinction angstrom exponent between 450-550nm, the
single scattering albedo at 550nm, the difference of single scattering albedo in two wavelengths between
450-550nm, the absorption coefficient at 550nm, the absorption angstrom exponent between 450-550nm,
and the real part of the refractive index at 532nm. We then use these aerosol optical properties, prescribe
a well-separated AMT training dataset and use the pre-specified clustering and Mahalanobis classification
method to derive optical-based AMTs during SEAC⁴RS. We find that by using any of sixteen
combinations of these six optical parameters, over 65% of optical-based wildfire biomass burning and
biogenic AMTs agree with their source-based analogue. We find that all four types studied (Biogenic, BB
from wildfires, BB from agricultural fires, and polluted dust) when prescribed using mostly airborne in
situ gas measurements, can be successfully extracted from at least three combinations of airborne *in situ*
aerosol optical properties over the US during SEAC⁴RS, such that more than 70% of optical observations
are typed consistently with source-based analog. However, we find that misclassifications are not evenly
distributed across the classes, and specifically the optically based classifications for BB from agricultural



fires and polluted dust include a large percentage of misclassifications that limit the usefulness of results relating to those classes.


## 5. Discussion

We suggest a similar study should be performed using data from additional airborne field campaigns which have the necessary, or equivalent, gas-phase measurements to derive source based-AMTs and many of the critical optical properties to extract optical based-AMTs. First, this would provide more robust statistics e.g., particular attention should be given to revisit the BB from agricultural fires and polluted dust AMTs in this study. Second, this would provide more AMTs/sub-AMTs to analyze e.g., Urban and Marine AMTs should be visited during CAMP²EX (Clouds, Aerosol and Monsoon Processes-Philippines Experiment) or KORUS-AQ (An International Cooperative Air Quality Field Study in Korea) and other types of BB and at different aging stages should be visited during FIREX-AQ (Fire Influence on Regional to Global Environments and Air Quality). Finally, this would also help assess if these chemical and optical signatures are reproducible from one year to another.

In this study, we obtained *in situ* aerosol optical signatures. Another essential step should be to examine optical signatures from space-based passive remote sensor(s), which derive total column effective ambient aerosol optical properties (instead of properties measured at the altitude of the aircraft in this study). One way to answer this question would be to compare the defined optical-based classes (DO-Class) signatures (i.e., means, variances and covariances that define the classes) using collocated airborne *in situ* aerosol optical properties and total column aerosol optical properties measured or inferred by sunphotometry (e.g., airborne 4STAR, Spectrometers for Sky-Scanning Sun-Tracking Atmospheric Research (Dunagan et al., 2013) or ground-based AERONET). This DO-Class database could then be used as a optical-based training dataset to enable widespread derivation of optical-based AMTs (DO-AMTs) using existing and future orbital and suborbital remote sensing instruments and networks.

The space mission addressing the designated observable Aerosol, Cloud, Convection and Precipitation (ACCP) from the NASA decadal survey (National Academies of Sciences, Engineering, and Medicine, 2018) is currently designing its suborbital (airborne and ground-based) component to address science questions that cannot be addressed from space (e.g., bridging satellite-inferred aerosol optical properties and aerosol speciation). This study illustrates how essential it is to explore existing airborne datasets to bridge chemical and optical signatures of different AMTs, before the implementation of future spaceborne missions and their corresponding suborbital field campaign(s) (e.g., upcoming spaceborne polarimeters SPEXone (Hasekamp et al., 2019) and Hyper-Angular Rainbow Polarimeter HARP-2 onboard the NASA Plankton, Aerosol, Cloud, ocean Ecosystem (PACE) (Werdell et al., 2019) and the multi-viewing multi-channel multi-polarization imager (3MI) (Fougnie et al., 2018) to be launched in the next 3 years or the next generation of Earth Observing System (EOS) satellites addressing NASA's ACCP).

…





Most of the six optical properties in this study (i.e., extinction angstrom exponent, single scattering albedo, difference of single scattering albedo, absorption coefficient, absorption angstrom exponent, and real part of the refractive index) are routinely derived by *in situ* and remote sensing instrumentation/networks (see Table 4). Some optical properties are more likely to present a higher uncertainty when measured from suborbital field campaigns and/ or from satellites. The real part of the refractive index, for example, although generally more uncertain, is highly desirable in many combinations of optical parameter to capture both the BB from wildfires and biogenic AMTs in this study. We strongly suggest future airborne campaigns consider including *in situ* measurements of AAE and RRI (very few of the campaigns to date flew PI-Neph and/or DASH-SP instruments) and a special attention should be given to deriving these parameters accurately from space. Our analysis has the advantage of providing alternate combinations of optical parameters when one optical parameter is either not available or too uncertain.

| | High | | Medium | | Low |

| **Aerosol Optical Parameter** | **Routinely Observed from Aircraft** | **Routinely Observed from Satellites** | **Importance as per this study** |
|---|---|---|---|
| Extinction Angstrom Exponent, EAE | 🟢 High | 🟢 High | 🟢 High |
| Single Scattering Albedo, SSA | 🟩 Medium | 🟩 Medium | 🟢 High |
| Difference in SSA, dSSA | 🟩 Medium | 🟩 Medium | 🟢 High |
| Absorption Coefficient, AC | 🟩 Medium | 🟧 Low | 🟢 High |
| Aerosol Absorption Exponent, AAE | 🟩 Medium | 🟩 Medium | 🟢 High |
| Real Refractive Index, RRI | 🟧 Low | 🟧 Low | 🟢 High |

**Table 4: Frequency at which the six aerosol optical parameters in our study are routinely derived from aircraft and current passive satellite sensors and importance of these optical parameters in our study. RRI: Real Refractive Index, AAE: Absorption Angstrom Exponent, AC: Absorption Coefficient, dSSA: difference in Single Scattering Albedo, SSA: Single Scattering Albedo, EAE: Extinction Angstrom Exponent**

Ultimately, this technique and its results has the potential to provide a much broader observational aerosol data set to evaluate global transport models than is currently available. Current satellite derived AMTs seem to marginally help models. One way to assess models would be to directly compare satellite derived





AMTs to AMTs derived from modeled optical properties (which are, in turn, computed from modeled chemical composition) using the same classification method (e.g., Taylor et al., 2015, Dawson et al. (2017), Nowottnick et al. (2015), Meskhidze et al. (2021)). However, it would be difficult to define the main source of errors in the case of a disagreement between model- and observation-based AMTs.
Potential causes of such a disagreement could be a combination of observation and method-specific errors or model-specific errors (e.g., the assumed model size distribution, dry refractive index, growth factor per specie, mass extinction efficiency per species, estimated mass per species, RH, transport, chemical processing, emissions, and other physiochemical variables). Let us emphasize that the technique and results in this study, alone, will not be able to fully explain any discrepancies between model and
observations. However, we suggest that the use of near-simultaneous gas-phase, chemical and optical instruments on the same aircraft restrict the causes of a disagreement between model- and observation-based AMTs to mostly model-specific errors. Moreover, as the AMTs in this study are less ambiguously defined (e.g., to each AMT corresponds an averaged distribution of aerosol chemical composition), we suggest that this may allow the assessment (and, by extension, improvement) of a few aerosol processes
simulated in CTMs.

## Appendix A

### A.1 Additional Information on Methods

### A.1.1 Method to Cloud-screen, Filter, and Humidify Airborne Observations

This section describes the cloud-screening, filtering, humidification, and colocation involved in the computation of the final set of sixteen optical parameters (i.e., EAE, dSSA and AAE between 450-550, 550-700 and 450-700 nm, Absorption coefficient, SSA at 450, 550 and 700nm and the RRI at 532 nm) in this study.

The LARGE TSI nephelometer and PSAP instruments operate under dry conditions. The only measurement provided at ambient conditions is the extinction coefficient at 532nm.
In this work, we need LARGE extinction and scattering coefficients at 450, 550 and 700nm at ambient conditions. To do that, we use the parameter "fRH550_RH20to80" at 550 nm provided by the LARGE f(RH) system (different from the TSI or PSAP instruments) and an exponential curve to obtain the impact of hygroscopic growth on the aerosol light scattering coefficient i.e., the scattering enhancement factor
f(RH) at 450, 550 and 700 nm. f(RH) is defined as the ratio of scattering coefficients in ambient over dry conditions. Ambient scattering at 550 nm, for example, is computed as the product of dry scattering at 550 nm and f(RH) at 550nm.

We filter out any values of LARGE dry scattering coefficient at 450 nm $\leq$ 10 Mm$^{-1}$ and LARGE ambient single scattering albedo coefficient (i.e., the ratio of scattering to extinction coefficient) at 863 nm $\leq$ 0.7.





Let us emphasize that, in this work, we use LARGE airborne *in situ* optical properties at the altitude of the aircraft (see first column of Table A1). However, we choose parameter names closer to what would be measured by remote sensing instruments, and which would represent a total or partial atmospheric column (see the second column of Table A1).

| In-situ optical parameters | What we call them in this study |
|---|---|
| Absorption Coefficient, AC | Absorption Coefficient, AC |
| Scattering Coefficient, SC | Scattering Coefficient, SC |
| Extinction Coefficient, EC | Extinction Coefficient, EC |
| Extinction Angstrom Coefficient, EAC | Extinction Angstrom Exponent, EAE |
| Scattering Angstrom Coefficient, SAC | Scattering Angstrom Exponent, SAE |
| Absorption Angstrom Coefficient, AAC | Absorption Angstrom Exponent, AAE |
| Single Scattering Albedo Coefficient, SSAC | Single Scattering Albedo, SSA |

**Table A1** *In-situ* optical parameters are provided at a given aircraft altitude in this study. The way we call these parameters is similar to what would be observed from remote sensing instruments.

The Extinction Angstrom Coefficient (EAC) between wavelengths $\lambda 1$ and $\lambda 2$ is computed using the Extinction Coefficient (EC) as follows:

$$EAC_{\lambda 1, \lambda 2} = \ln(EC_{\lambda 2} - EC_{\lambda 1}) / \ln(\lambda 1) - \ln(\lambda 2) \qquad \text{(Eq. A.1.1a)}$$

The Single Scattering Albedo Coefficient (SSAC) at wavelength $\lambda 1$ is computed using the Scattering Coefficient (SC) and the Extinction Coefficient (EC) at $\lambda 1$ as follows:

$$SSAC_{\lambda 1} = SC_{\lambda 1} / EC_{\lambda 1} \qquad \text{(Eq. A.1.1.b)}$$

The Absorption Angstrom Coefficient (AAC) between wavelengths $\lambda 1$ and $\lambda 2$ is computed using the Absorption Coefficient (AC) as follows:

$$AAC_{\lambda 1, \lambda 2} = \ln(AC_{\lambda 2} - AC_{\lambda 1}) / \ln(\lambda 1) - \ln(\lambda 2) \qquad \text{(Eq. A.1.1.c)}$$

And the Absorption Coefficient (AC) at wavelength $\lambda 1$ is computed as follows:

$$AC = EC_{\lambda 1} - SC_{\lambda 1} \qquad \text{(Eq. A.1.1.d)}$$

DASH-SP provides measurements of Real Refractive Index at 532 nm (RRI), $RRI_{DASH-SP\_dry}$, information on the particle hygroscopicity, $\kappa_{DASH-SP\_dry}$, and the particle diameter, $Dp_{DASH-SP\_dry}$, in dry conditions. We compute DASH-SP RRI in ambient conditions, $RRI_{DASH-SP\_ambient}$, using $RRI_{DASH-SP\_dry}$, $\kappa_{DASH-SP\_dry}$, and the ambient relative humidity and temperature measurements, $RH_{HSKP}$ and $T_{HSKP}$, provided by the AIMMS-20 or 3D-winds instruments. First, we vary the Growth Factor, $GF_{var}$, from 1.02 to 1.5 by increments of 0.01 and compute the particle hygroscopicity, $\kappa_{var}$, for given $RH_{HSKP}$, $T_{HSKP}$ and $Dp_{DASH-SP\_dry}$ measurements as follows:

$$\kappa_{var} = (GF_{var}{}^3 - 1) \times (1 - \kappa_a) / \kappa_a \qquad \text{(Eq. A.1.1e)}$$

Where:





- $\kappa_a = (RH_{HSKP} \, / \, 100\%) \, / \, \exp(C_{amb} \, / \, ( \, GF_{var} \, x \, Dp_{DASH\text{-}SP\_dry}))$
- $C_{amb} = (4 \, x \, \sigma_{sa} \, x \, M_w) \, / \, (R \, x \, T_{HSKP} \, x \, \rho_w)$
- $\sigma_{sa} = 0.0761\text{-}1.55 \, x \, 1e\text{-}4 \, x \, (T_{HSKP} \, \text{-}273);$
- $M_w = 18.01528/1000$ kg/mole
- $R = 8.3144598$
- $\rho_w = 1000$ kg/m$^3$

We select the growth factor, $GF_{var}$, that provides the closest $\kappa_{var}$ value to the $\kappa_{DASH\text{-}SP\_dry}$ measurement. We call this growth factor $GF_{select}$. Finally, we compute the ambient RRI, $RRI_{DASH\text{-}SP\_ambient}$, using $RRI_{DASH\text{-}SP\_dry}$ and $GF_{select}$ obtained in the precious steps and equation 5 of Mallet et al. (2003) (based on Hänel (1976)) as follows:

$$RRI_{DASH\text{-}SP\_ambient} = RRI_w + (RRI_{DASH\text{-}SP\_dry} - RRI_w) \, x \, (GF_{select})^{-3} \qquad \text{(Eq. A.1.1f)}$$
Where $RRI_w = 1.33$

Let us note that Aldhaif et al. (2018) demonstrate the limitations of using the volume-weighted mixing rule approach above, especially in the presence of OA.

The PI-Neph provides measurements of dry phase function ($P_{11}$) and the second element of the scattering phase matrix ($P_{12}$) at three wavelengths over an angular range spanning >170°. These measurements are fed into the GRASP (Dubovik et al., 2014)) algorithm to obtain retrieved values of spectral complex refractive index, a parameterized size distribution as well as derived optical properties like scattering coefficients. In this work we utilize these optical properties provided by PI-Neph in dry conditions: the scattering coefficients at 532 nm, $scat_{PI\text{-}Neph\_dry}$, the dry size distribution, $dNdlnr_{PI\text{-}Neph\_dry}$ and the Refractive Index, $RI_{PI\text{-}Neph\_dry}$, composed of a real (RRI) and Imaginary part (IRI). First, we compute the "target" ambient scattering coefficient at 532 nm, $scat_{PI\text{-}Neph\_target}$, as the product of $scat_{PI\text{-}Neph\_dry}$ and LARGE f(RH) measurements at 550nm. Second, we compute the ambient scattering coefficient at 532 nm, $scat_{PI\text{-}Neph\_ambient}$, corresponding to each $GF_{var}$ from 1 to 1.5 by increments of 0.01 using (i) a Mie code (Mishchenko et al., 2002) and, as input to the Mie code, (ii) the ambient size distribution and corresponding radii, computed from $dNdlnr_{PI\text{-}Neph\_dry}$ and $GF_{var}$, (iii) the ambient refractive index computed from $RI_{PI\text{-}Neph\_dry}$ and $GF_{var}$ (see Eq. A.1.1f) and a prescribed geometric standard deviation (i.e., ~1.12, which results in similar computed and provided $scat_{PI\text{-}Neph\_dry}$ values when using the same Mie code and initial parameters $dNdlnr_{PI\text{-}Neph\_dry}$ and $RI_{PI\text{-}Neph\_dry}$). Third, we select $GF_{var}$ (we call this growth factor, $GF_{select}$) and corresponding $RRI_{PI\text{-}Neph\_ambient}$ that records the minimum difference between $scat_{PI\text{-}Neph\_ambient}$ and $scat_{PI\text{-}Neph\_target}$.

We compute ambient AMS and SP2 mass concentrations using the parameter "stdPT-to-AMB_Conversion_AMS-60s" reported with the AMS data. SP2 Black Carbon (BC) standard concentration (referred to as "refractory black carbon", and experimentally equivalent to elemental carbon at the 15% level (Petzold et al., 2013; Kondo et al., 2011; Perring et al., 2017)), originally in ng.m$^{-3}$, is converted into µg.m$^{-3}$ and scaled upwards, on a flight-by-flight basis, to represent the entire accumulation





mode (on average by 1.14). The AMS sulfate, ammonium and nitrate are normalized to the sum of sulfate, ammonium, and nitrate (see first row of Fig. 4c). The AMS OA and SP2 BC are normalized to the sum of OA, BC, sulfate, ammonium, and nitrate (see second row of Fig. 4c).

In the case of SAGA, bromide and chloride are set to zero if under the detection limit of 0.0107 and 0.0391 µg.m$^3$.

In the case of PALMS, we use volume weighted products (Froyd et al., 2019). In this study, PALMS particle classes include mineral dust, sea salt, biomass burning, and sulfate-organic-nitrate mixtures (SON). The SON class was further refined into organic-rich, sulfate-rich, and nitrate-rich particle types,

plus a remainder of SON particles that did not exhibit a dominant chemical sub-component. To define the Marine and Polluted Dust AMTs, PALMS composition was combined with aerosol size distribution data from LARGE to yield integrated volume fractions of mineral dust and sea-salt particle types from D=0.1-5 µm based on the method of Froyd et al. (2019). In Fig. 4 the average AMT chemical composition is determined as a raw number fraction of particles observed by PALMS.


## A.1.2 Method to Collocate Airborne Observations

All the airborne observations are cloud screened using wing-mounted cloud probes. Table A2 defines three datasets used in this study with its associated number of data points, called AIRBO$_1$, AIRBO$_2$ and AIRBO$_3$ and their combination, AIRBO. In the AIRBO$_{1,2,3}$ and ultimately the AIRBO datasets, the

LARGE data is first collocated to housekeeping (HSKP) data (i.e., select same "start_utc" in seconds) and humidified/ filtered (see A.1.1).

In the AIRBO$_1$ dataset, we compute the mean HSKP and LARGE values in a ± 30 second range centered on each collocated AMS-PALMS-SP2 "start_time" (i.e., the 1 min "merged" file). We then record

LARGE averaged values if (i) the average is made of at least 20 points and (ii) the standard deviation of the LARGE EAE is below 30%.

In the AIRBO$_2$ dataset, we compute the mean HSKP and LARGE values between each DASH-SP "start_utc" and "end_utc". DASH-SP measurements are then filtered and humidified using the dry

DASH-SP and the averaged ambient HSKP measurements (see A.1.1). We record HSKP, LARGE and DASH-SP values if the following four parameters are below 30%: (i) the standard deviation of the LARGE EAE, (ii) the difference between $\kappa_{\text{DASH-SP\_dry}}$ and $\kappa_{\text{var}}$ (see A.1.1), (iii) the standard deviation of RH$_{\text{HSKP}}$ and (iv) the standard deviation of T$_{\text{HSKP.}}$

In the AIRBO$_3$ dataset, we compute the mean HSKP and LARGE values between each PI-Neph "start_utc" and "end_utc". PI-Neph measurements are then filtered and humidified using LARGE dry scattering coefficient and f(RH) measurements (see section 6.1.3). We record HSKP, LARGE and PI-Neph values if the following four parameters are below 30%: (i) the standard deviation of the LARGE EAE, (ii) the standard deviation on scat$_{\text{PI-Neph\_dry}}$, (iii) the standard deviation on LARGE f(RH), and the

difference between PI-Neph scat$_{\text{PI-Neph\_target}}$ and scat$_{\text{PI-Neph\_ambient}}$.





Finally, we collocate the HSKP-LARGE-DASH-SP (HSKP-LARGE-PI-Neph) to the AMS-PALMS-SP2 datasets in the case of AIRBO$_2$ (AIRBO$_3$). To do that, if there are multiple AMS-PALMS-SP2 data points between each HSKP-LARGE-DASH-SP (HSKP-LARGE-PI-Neph) averaged time stamp, we average all
AMS-PALMS-SP2 data between the HSKP-LARGE-DASH-SP (HSKP-LARGE-PI-Neph) averaged time stamps. If there are no multiple AMS-PALMS-SP2 data points between the HSKP-LARGE-DASH-SP (HSKP-LARGE-PI-Neph) averaged time stamps, we select the closest AMS-PALMS-SP2 data in time to the HSKP-LARGE-DASH-SP (HSKP-LARGE-PI-Neph) averaged time stamps.

The dataset in this study, AIRBO, was obtained by first, selecting common 1 min UTC time stamps from all 3 datasets, and then arbitrarily selecting, in order of priority when present, AIRBO$_2$, AIRBO$_1$ and AIRBO$_3$.

| Name of Dataset | Instruments (see Table 1) | Temporally Co-located Aerosol Optical Parameters (see Table 1, 2) | Valid Number of Data Points (*) |
|---|---|---|---|
| AIRBO$_1$ | LARGE | EAE, SSA, dSSA, AC, AAE | 871 |
| AIRBO$_2$ | LARGE, DASH-SP | EAE, SSA, dSSA, AC, AAE, RRI | 716 |
| AIRBO$_3$ | LARGE, PI-Neph | EAE, SSA, dSSA, AC, AAE, RRI | 176 |
| AIRBO (This study) | LARGE, DASH-SP, PI-Neph | EAE, SSA, dSSA, AC, AAE, RRI | 781 |

**Table A2: Definition of three datasets (AIRBO$_1$, AIRBO$_2$, AIRBO$_3$) and their combination, AIRBO**
**(which is the dataset used in this study), the airborne instruments involved during SEAC$^4$RS, the co-located parameters (EAE: Extinction Angstrom Exponent, SSA: Single Scattering Albedo, dSSA: difference in SSA at 2 wavelengths, AC: absorption Coefficient and AAE: Absorption Angstrom Exponent and RRI: Real Refractive Index) and the (*) number of data points showing valid aerosol optical properties and one valid PS-AMT (among BBAg., BBWild., Bio. or PollDust)**


**A.1.3 Method to Define Prescribed Source-based AMTs (PS-AMTs)**

This section explains Fig. 2 (and step 1 in Fig. 1) in more details. We refer the reader to Table 1 for all the instruments in this section. First, we define Polluted Dust PS-AMT using PALMS dust number fraction (i.e., PALMS 'MineralFrac_PALMS') above 0.15 and integrated the Dry Aerosol Volume
Concentration by the APS (i.e., 'IntegV_Daero-PSL_APS_LARGE' above $2\mu m^3 cm^{-3}$; note that APS measurements sampled dry aerodynamic diameters ranging from 0.56 to 6.31 μm (Espinosa et al., 2018)). Similarly, we define Marine PS-AMTs when PALMS sea-salt number fraction > 0.15 and total volume >2 μm$^3$ cm$^{-3}$. The remaining observations may then be evaluated for Biomass Burning (BB) PS-AMTs if gas-phase measurements of acetonitrile, isoprene, monoterpene and CO (i.e., using PTRMS
"Acetonitrile", WAS "Isoprene_WAS", PTRMS "Isoprene-Furan", PTRMS "Monoterpenes", WAS "CO_WAS" and DACOM "CO_DACOM") meet certain prespecified thresholds. More specifically,





observations are classified as BB if (i) acetonitrile > 250 x $10^{-3}$ ppbv, or (ii) (acetonitrile > 190 x $10^{-3}$ ppbv) & (acetonitrile/(isoprene + monoterpene)>2.5) or (iii) CO>250 ppbv. BB PS-AMTs are further differentiated as coming from prescribed (agricultural) fires (called "BB Ag.") if the longitude is east of
-95º or from wildfires (called "BB wild.") if the longitude is west of -95º.The -95º longitude threshold was selected according to Fig. 3 and the location of Ag. fires (green triangles) according to Liu et al. (2016). If observations are not classified as dust or BB, we classify them as biogenic if isoprene + monoterpene > 2ppbv. Finally, remaining observations are classified as urban if the altitude is below 3km and NO2 > 1 ppbv (i.e., using the NOAA Nitrogen Oxides and Ozone ($NO_y/O_3$ $NO_2$ "NO2_ESRL" or
TDLIF "NO2_TDLIF").

**A.1.4 Method to Select Most Useful and Well Separated Aerosol Optical Properties**

This section explains step 2 of Fig. 1 in more details. Figure A1 is a simplified example to illustrate our method. It shows only two optical parameters (i.e., SSA and EAE) and three hypothetical PS-AMTs (e.g.,
"pure" dust in red, marine in blue and BB in green) measured by one hypothetical optical instrument in two different environments (defined by different locations and times, Fig. A1 a-b vs. A1 c-d). Fig A1 a-b shows a smaller hypothetical range of EAE and SSA for the BB PS-AMT (green cluster), compared to Fig. A1 c-d.
To answer the questions "how well are these PS-AMTs (i.e., red, blue and green clusters in either Fig. A1
a or A1 c) separated" i.e., "are the optical signatures of these PS-AMTs distinct?", we (i) select each data point separately (e.g., yellow crosses on Fig A1 b and A1 d), (ii) recompute each PS-AMT cluster with the data point excluded (i.e., different blue PS-AMT on A1 b and green PS-AMT on A1 d compared to A1 a and A1 c) and calculate the Mahalanobis distance (Mahalanobis, 1936; Burton et al., 2012). The Mahalanobis distance is the distance between the data point in question (i.e., yellow crosses on Fig. A1 b
or A1 d) and the position of each cluster center (i.e., red, blue, and green clusters on Fig. A1 b or A1 d), which depends on cluster center, tilt and width in a multi-parameter space. These distances are called $D_1$, $D_2$ and $D_3$ on either Fig. A1 b or A1 d. In the case of the yellow cross on Fig. A1 b, distance $D_1$ is the smallest and the test point is reassigned to its original cluster. The test point is by consequence well separated from other clusters and "steady". On the other hand, distance $D_1$ is also the smallest on Fig. A1
d, which means the test point (yellow cross) on Fig. A1 d is not reassigned to its original cluster. The test point is by consequence not well separated from other clusters in this case and not "steady". "Steady fraction" is the fraction of cases within each PS-AMT that are correctly identified. "Steady" fractions are used to assess separation between PS-AMTs. When including additional components (e.g., any other aerosol optical parameter from Table 2 in addition to SSA and EAE on Fig. A1), the additional number
of "steady points" shows the component's relative importance in separating the PS-AMTs. The yellow points that are "steady" on Figure A1 (i.e., correctly classified, or well separated) are used to define the most useful and well separated aerosol optical properties for each PS-AMT (i.e., step 2 in Fig. 2).





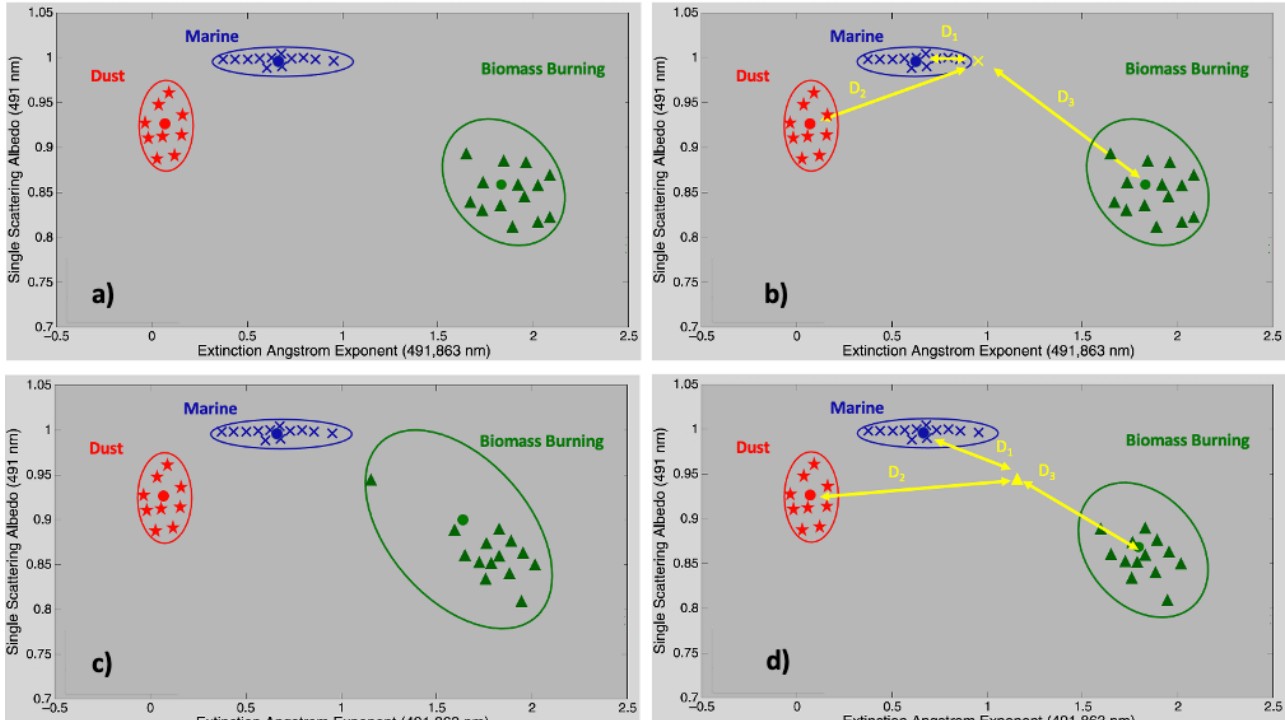

**Figure A1: Conceptual/ hypothetical illustration of how we quantify separation between different air mass types, select the most useful and well separated aerosol optical parameters. It shows three hypothetical PS-AMTs (e.g., dust in red, marine in blue and BB in green) measured by one hypothetical optical instrument (Fig. A1 a-b) in one environment and another (Fig. A1 c-d). The EAE and SSA values in this illustration are based on AERONET observations (Russell et al, 2014)**
**and are representative of typical "pure" dust, marine and BB total column remote sensing inferred ground based EAE and SSA values. Note that it only shows two dimensions even though some calculations of Mahalanobis Distances (e.g., $D_1$, $D_2$, $D_3$) will be made using more dimensions in this study.**

**A.2 Additional Information on Results**

**A.2.1 Aerosol Optical Parameters classified by PS-AMT**

This section describes the ranges of the sixteen aerosol optical parameters (i.e., EAE, SSA, dSSA, AAE and AC at different combinations of 450, 550 and 700 nm and RRI at 532 nm from Table 2), classified by PS-AMTs in our study.






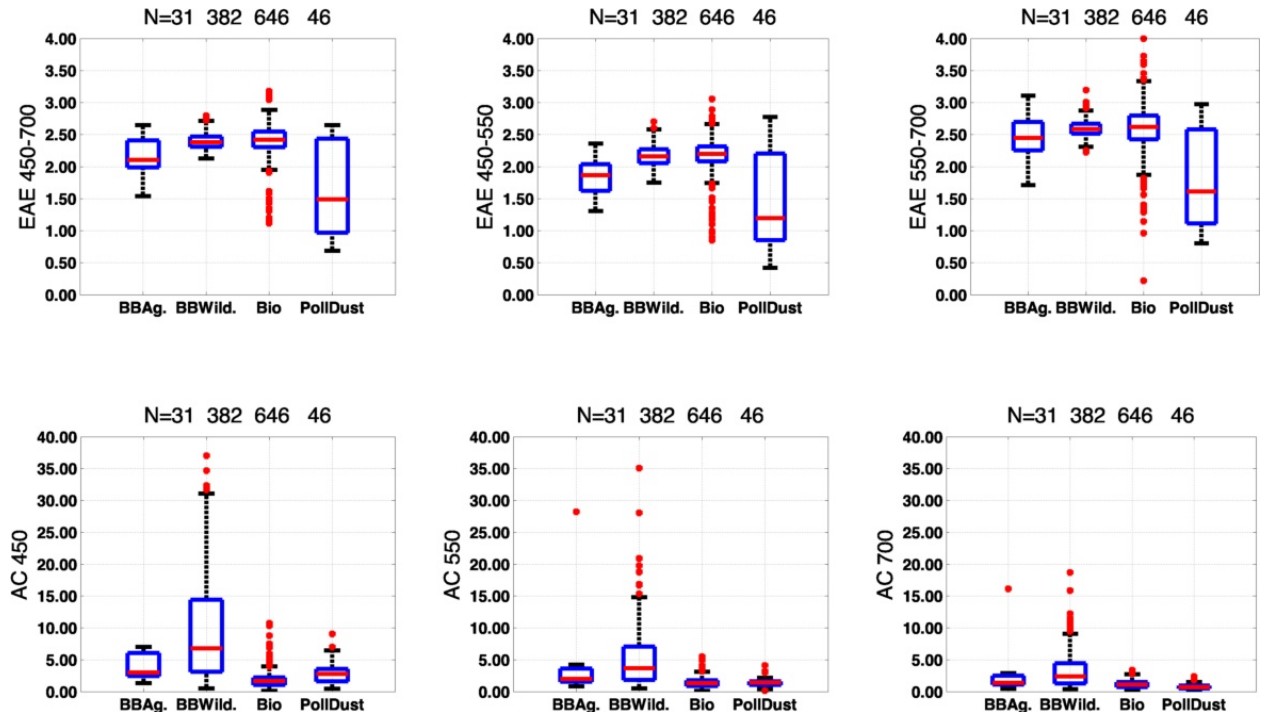

**Figure A2: EAE (450-700nm, 450-550nm, 550-700nm) and AC (450, 550 and 700nm) per PS-AMTs.
In each blue box, the red horizontal line indicates the median, and the bottom and top edges of the
box indicate the 25th and 75th percentiles, respectively. The black whiskers extend to the most
extreme data points not considered outliers, and the outliers are plotted individually using red
points. Let us note that the LARGE EC measurements at 700 nm experienced issues during the
latter half of SEAC4RS (Shinozuka et al., *pers. comm.*). AC: Absorption Coefficient, EAE:
Extinction Angstrom Exponent. Numbers in the title correspond to the number of points behind
each box-whisker for the respective BBAg., BBWild., Bio. and PollDust PS-AMTs.**





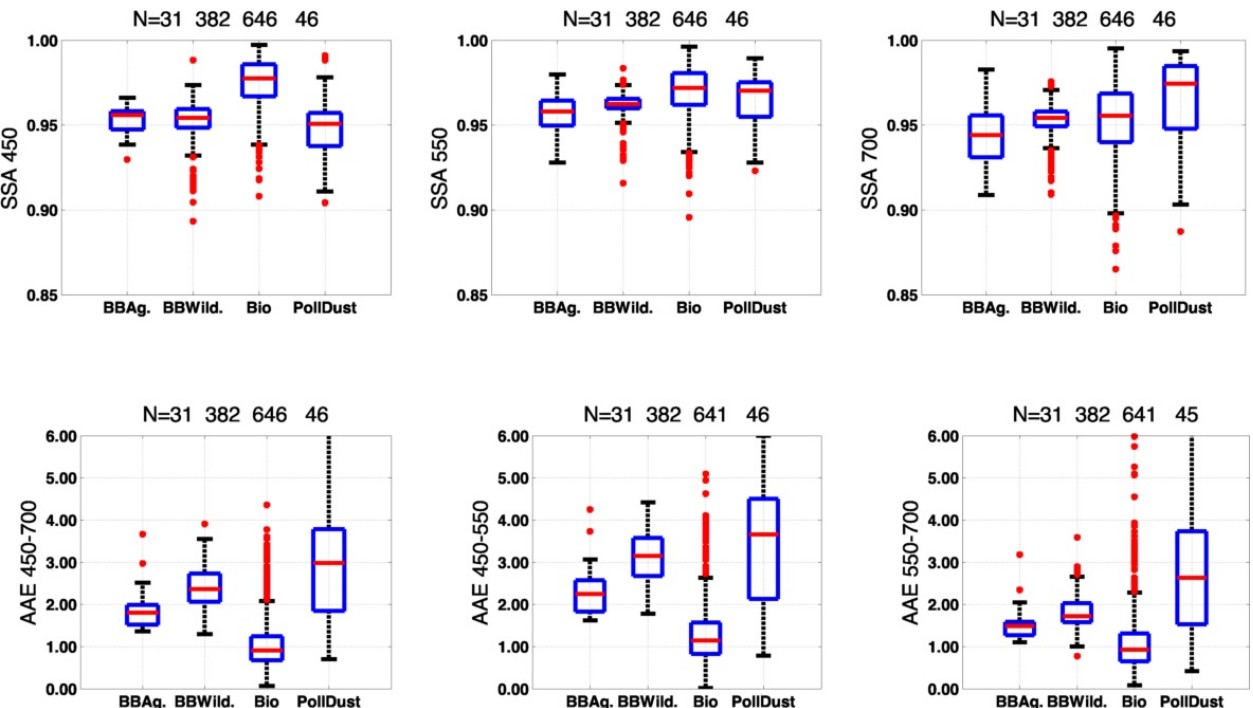

**Figure A3: SSA (450, 550 and 700nm) and AAE (450-700nm, 450-550nm, 550-700nm) per PS-AMTs. In each blue box, the red horizontal line indicates the median, and the bottom and top edges of the box indicate the 25th and 75th percentiles, respectively. The black whiskers extend to the most extreme data points not considered outliers, and the outliers are plotted individually using red points. AAE: Absorption Angstrom Exponent, SSA: Single Scattering Albedo. Numbers in the title correspond to the number of points behind each box-whisker for the respective BBAg., BBWild., Bio. and PollDust PS-AMTs.**



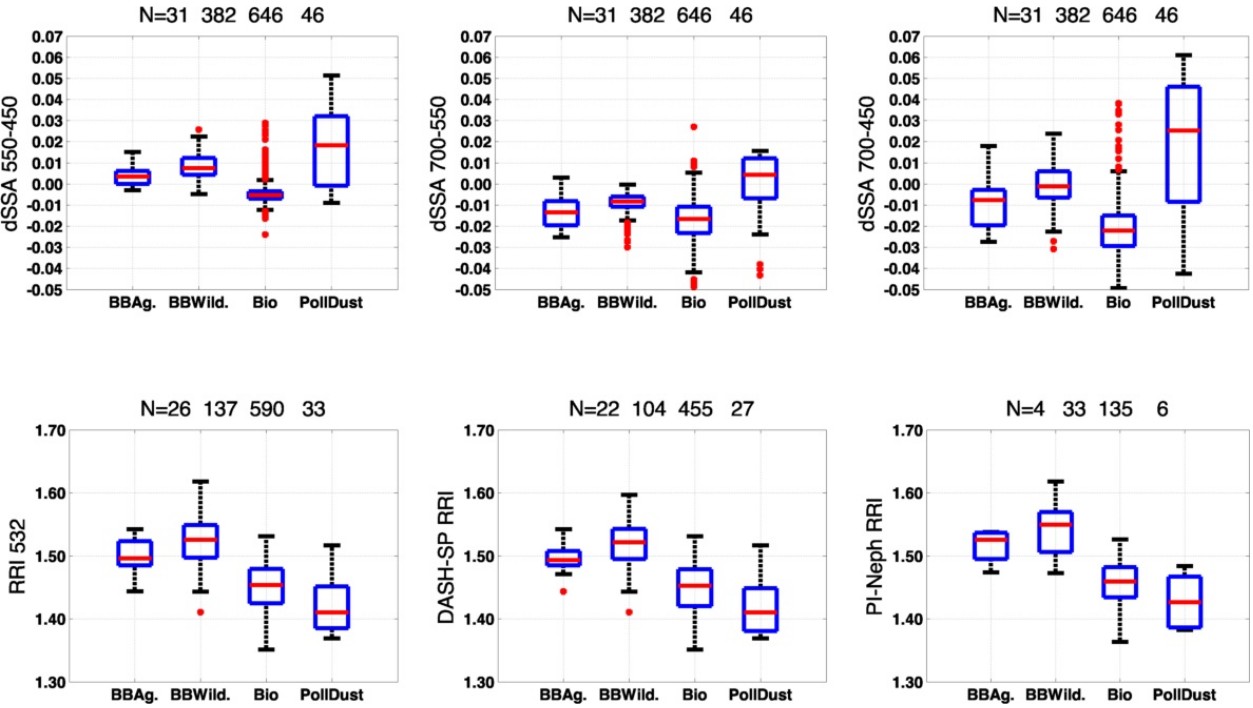


**Figure A4: dSSA (700-450nm, 550-450nm, 700-550nm), RRI (from DASH-SP and PI-Neph), RRI from DASH-SP and RRI from PI-Neph at 532 nm per PS-AMTs. In each blue box, the red horizontal line indicates the median, and the bottom and top edges of the box indicate the 25th and 75th percentiles, respectively. The black whiskers extend to the most extreme data points not**
**considered outliers, and the outliers are plotted individually using red points. RRI: Real Refractive Index, dSSA: difference in Single Scattering Albedo. Numbers in the title correspond to the number of points behind each box-whisker for the respective BBAg., BBWild., Bio. and PollDust PS-AMTs.**

Note the slightly lower RRI values for DASH-SP, compared to PI-Neph (i.e., respectively 1.41 and 1.43
at 532nm in Fig. A4) in the case of PollDust PS-AMTs. We explain this difference in RRI values by different PollDust PS-AMT Growth Factor (GF) values. We obtain GF through two methods: (1) the values directly measured by DASH-SP for particles in the size range $0.18 < d_{dry} < 0.40$ μm and (2) through an iterative procedure matching the output of a Mie code with dry PI-Neph retrievals and f(RH) measurements made by the LARGE group in parallel (see section A.1.1 for more details). We find a
respective median PollDust PS-AMT GF value of ~1.3 and ~1.2 in the case of DASH-SP and PI-Neph, which we suggest is due to a smaller sampling size range for DASH-SP, compared to PI-Neph (see Table 1).





## A.2.2 Most Useful and Well Separated Aerosol Optical Properties – Sixteen Parameters

This section describes the percentage of points unambiguously retrieved or "steady" (i.e., points that are well separated from other clusters and, hence, reassigned to their initial clusters) when using different combinations of respectively two and three out of sixteen aerosol optical parameters across all four principal PS-AMTs (i.e., provides more details to section 3.2).


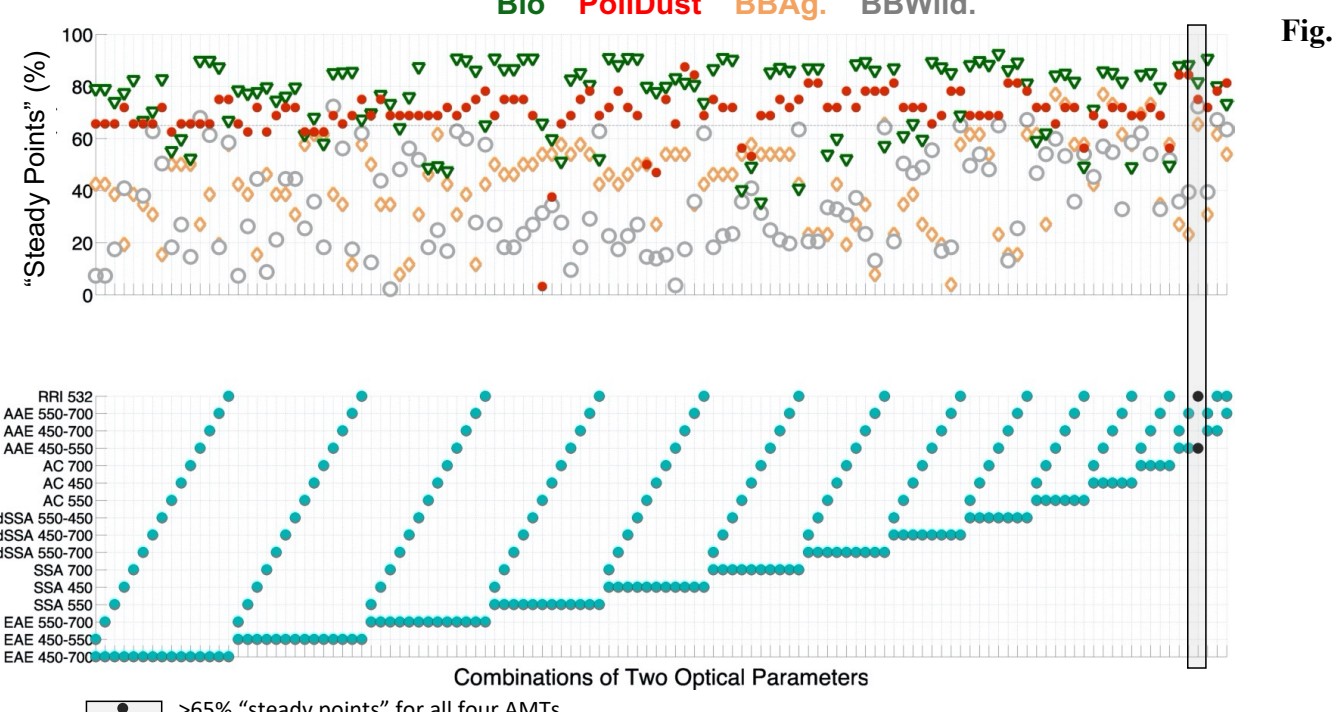

Fig.

**Figure A5: Percentage of points "steady" (i.e., fraction of cases of a given type that are correctly identified; see section 2.4 for more info) in the upper panel when using different combinations of**
**two aerosol optical parameters in the lower panel for each PS-AMT. Grey box and black points are combinations of optical parameters showing > 65% "steady" for PS-AMTs BBAg., BBWild., Bio.and PollDust. RRI: Real Refractive Index, AAE: Absorption Angstrom Exponent, AC: Absorption Coefficient, dSSA: difference in Single Scattering Albedo, SSA: Single Scattering Albedo, EAE: Extinction Angstrom Exponent.**






Fig.

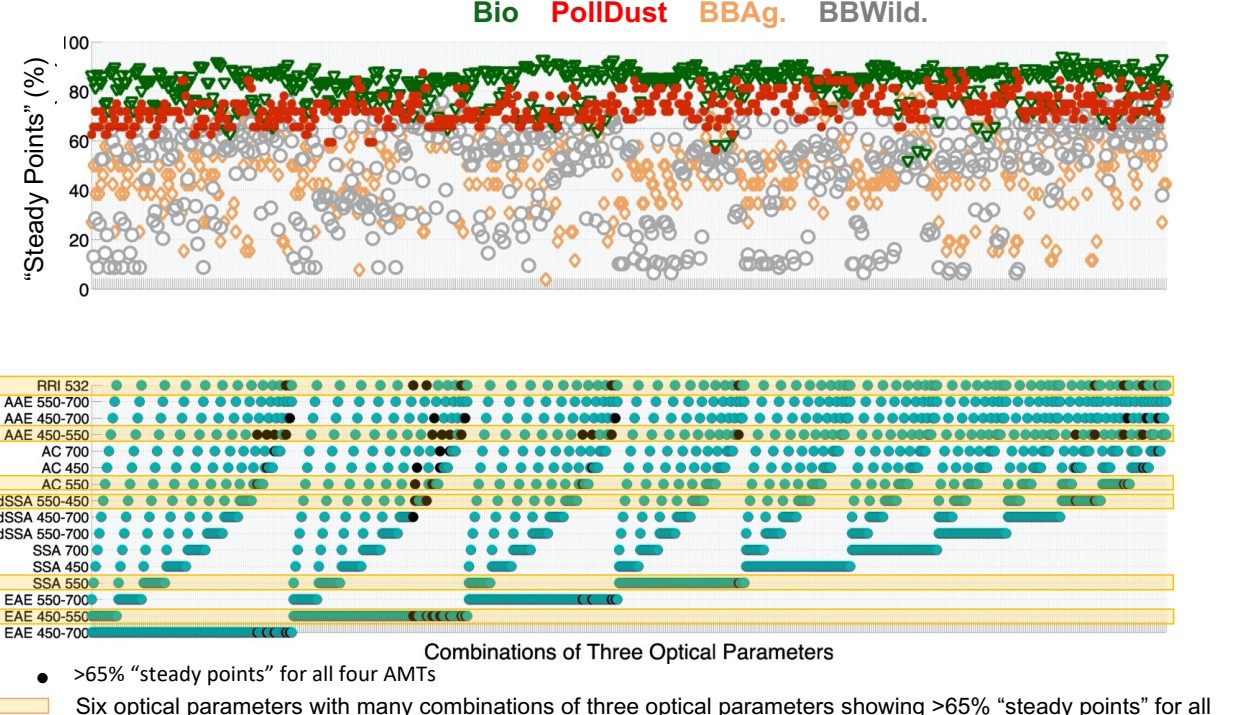

- ● >65% "steady points" for all four AMTs

- ▭ Six optical parameters with many combinations of three optical parameters showing >65% "steady points" for all four AMTs

**Figure A6: Percentage of points "steady" (i.e., fraction of cases of a given type that are correctly identified; see section 2.4 for more info) in the upper panel when using different combinations of**
**three aerosol optical parameters in the lower panel for each PS-AMT. Black points are combinations of optical parameters showing > 65% "steady points" for PS-AMTs BBAg., BBWild., Bio. and PollDust. RRI: Real Refractive Index, AAE: Absorption Angstrom Exponent, AC: Absorption Coefficient, dSSA: difference in Single Scattering Albedo, SSA: Single Scattering Albedo, EAE: Extinction Angstrom Exponent. Horizontal orange boxes show the selection of our**
**six aerosol optical parameters. Orange boxes show the six aerosol optical parameters that we have selected in the remainder of the study.**

### A.2.3 Composition of our Polluted Dust (PollDust) PS-AMT

This section shows a compositional picture of the PollDust PS-AMTs from PALMS (see Fig. A7). The
accumulation mode is a mixture of particle types, all of which contain sulfate and organic material. Coarse mode dust particles account for most of the aerosol volume, whereas a non-dust accumulation mode contributes most to the total number concentration of particles.






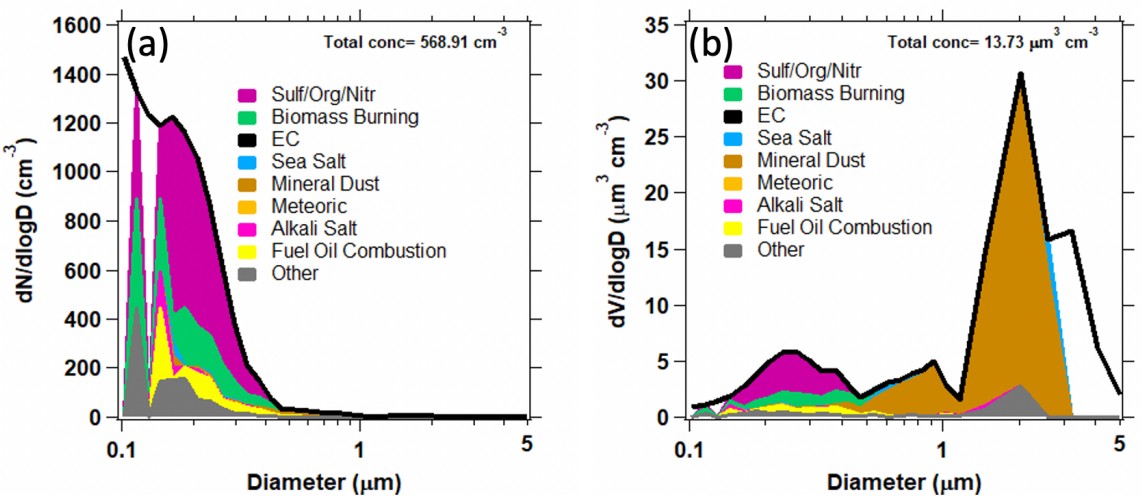

**Figure A7: PALMS particle classes are mapped to the total number (a) and volume (b) size distribution from LARGE based on the method of Froyd et al. (2019). Data include flight segments representative of the Polluted Dust PS-AMT.**


### Data Availability

The SEAC⁴RS data used in this study are publicly available at the following link: https://www-air.larc.nasa.gov/cgi-bin/ArcView/seac4rs?DC8=#top

### Author Contribution

The overarching research goals were formulated by MSK. Many co-authors influenced the evolution of these research goals. MSK and QT carried out the formal analyses. MSK carried out the investigations and visualizations and wrote the original draft. All coauthors have reviewed and edited the multiple drafts of the paper. The methodology behind the SCMC method was first developed by SPB and adapted to *in* 035 *situ* data by MSK.

### Competing Interests

The authors declare that they have no conflict of interest, except for Dr Armin Sorooshian being an editor of ACP.





## Acknowledgements

This research was supported by the NASA Atmospheric Composition Modeling and Analysis Program (ACMAP) under grant NNH14ZDA001N-ACMAP. We thank Dr R. Eckmann for his support. We appreciate the efforts of all the SEAC⁴RS *in situ* instrument principal investigators involved in this study for obtaining, processing, documenting, and disseminating their respective datasets. We also appreciate the comments of the reviewers that have helped us to improve this article.

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
