# Peer review of "Identifying Chemical Aerosol Signatures using Optical Suborbital Observations: How much can optical properties tell us about aerosol composition?"

_Atmospheric Chemistry and Physics, 2021_

## Author Response (AR1)

We thank both anonymous reviewer #1 (AR1) and reviewer #2 (AR2) for reviewing our manuscript. Their inputs are very similar, and we have made significant revisions to address them. We believe this has considerably strengthened our paper.

Author Comments to Anonymous reviewer #1:

AR1: The authors address the important question of how to reconcile the various definitions of aerosol types used in different communities that deal with in-situ measurements, remote-sensing observations, and modelling studies. In essence, the authors describe a methodology for inferring a chemically defined aerosol species based on measurements of optical parameters. A data set of in-situ measurements collected during an airborne campaign over the US forms the foundation of this study. The topic is highly relevant as such studies are needed to overcome the divide between the different communities that study atmospheric aerosols. The applied methods are sound but the presentation of the work leaves much to be desired as outlined more specifically below. Ultimately, the developed approach should be applicable to mostly passive remote-sensing observations from space. While this is touched upon in the discussion, the work would be much stronger if the authors were to present more details of how the findings could be used to exloit spaceborne observations. Those observations of column optical properties are less likely to meet the unmixed aerosol conditions used in this study. A way forward could be to mix different chemically defined AMTs, retrieve their resulting optical properties, and check if the retrieval would still be able to disentangle the contributions.

We propose a way to examine optical signatures from space-based passive remote sensor(s) in the discussion section. We also acknowledge that this study constitutes a first step towards the goal of translating the space-based "total atmospheric column effective" aerosol optical properties and derived optical-based AMTs into source-based AMTs. Many steps remain and we believe adding any text beyond what is currently written in our discussion would be outside the scope of this paper.

AR1: Overall, the study requires substantial revisions before it can be accepted for publication. And even then, it would fit much better within the scope of AMT rather than ACP, as it is focused on describing the methodology rather then applying it. This reviewer therefore recommends to move the contribution to AMT with major revisions rather than rejecting it for publication in ACP.

We have made significant revisions as detailed in the subsequent responses. Though we acknowledge that AMT might also be a good fit for our paper, we believe it fits the scope of ACP (i.e., studies with general implications for atmospheric chemistry and physics). This is because it attempts to bridge the gap between two often distinct atmospheric observational research fields (i.e., chemistry and physics) by assessing how much aerosol optical properties can tell us about aerosol composition. In this paper, we describe a methodology and then, apply it. The results have the potential to provide a much broader observational aerosol data set to evaluate global transport models than is currently available. It is also timely as it illustrates how essential it is to explore existing airborne datasets to bridge chemical and optical signatures of different aerosol types before the implementation of future spaceborne missions. This paper also suggests which aerosol optical property should be accurately measured/derived from future airborne field campaigns or space-borne sensors to better characterize the aerosols.

AR1: Here are some comments regarding the presentation. Once these items have been addressed, it will be much easier to assess the scientific quality of the work: The paper reads like a mash-up of an internal report and a PowerPoint presentation. Particularly the introduction reads like an aggregation of agency jargon.

By "agency jargon", we suspect the reviewer refers to the name of instruments and/ or missions. We find these necessary to keep in the text as it is customary in many peer-reviewed publications. We have added a list of acronyms and abbreviations to this paper (Appendix B) and we have deleted some acronyms when they were found not essential.

AR1: The structure is set up in a way that keeps the reader browsing back and forth to match the thoughts of the authors.

We especially thank the reviewer for this input. We have restructured the paper and believe it now presents a better flow. These are the many ways we have addressed this input: (i) the introduction was shortened and simplified, (ii) the method section was clarified, reduced and now contains parts of the former appendix, (iii) the instrumentation and data section now occurs before the method section, (iv) Table 2 now includes information from the appendix, (v) the appendix A was also clarified, consolidated and reduced and (vi) we have limited the references to sections and figures in the main text.

AR1: The naming of the investigated parameters (e.g. PS-AMT, DO-AMT, DO-Class) is unintuitive and confusing. The meaning of those acronyms is introduced very late in the manuscript and only indirectly in the form of section headings.

These parameters are now clearly defined in section 2, emphasized in a remodeled Figure 1 and present in the new list of abbreviations in appendix B.

AR1: Important information, such as the definition of the PS-AMTs (which should be part of the methodology) is scattered over Sections 2, 3, and the Appendix. The threshold values used in the categorization should be clearly defined and justified.

This was addressed by entirely restructuring section 2 and by including information from the appendix in the main text.

AR1: Figures are of bad quality or designed badly. Personally, this reviewer would prefer proper flow charts rather than the PowerPoint-style presentation in Figures 1 and 2.

We have remodeled Figure 1 and Figure 2 (as well as Figure 6 and Table 4) and believe they are now clearer and more esthetically pleasing.

AR1: The referencing to figures and tables is excessive and keeps the reader browsing to keep up with the text.

We have limited these references throughout the manuscript and slightly modified Figure 6 accordingly.

AR1: The text is littered with redundant statements in parantheses that re-state what has just been explained. Please pick one formulation and go with it.

We have limited the redundant statements throughout the manuscript.

AR1: Please double-check all equations. At least Eqs. (A.1.1a) and (A.1.1c) are not correct.

We have corrected these equations. Note that these were typos within the text and our calculations remain sound. These equations are now consolidated in Table 2 (instead of the appendix).

Author Comments to Anonymous reviewer #2:

AR2: The authors identified different species of chemical air mass aerosol types by using measurements of aerosol optical parameters. This study is overall well designed and the results are well presented. However, this manuscript looks more like a technical report, especially the abstract, introduction, and way to present the results. I agree with reviewer 1 that this article is more suitable for AMT instead of ACP. I suggest transferring the paper to AMT and making major changes before it can be accepted for publication. The authors are suggested to rewrite the Introduction and Method in a more professional way instead of listing everything one by one. Some paragraphs can be merged.

As previously mentioned in our responses to AR1, we have made considerable modifications to our manuscript (e.g., the introduction was shortened/ simplified, the method section was clarified, reduced and now contains parts of the former appendix, the instrumentation and data section now occurs before the method section, and we have limited the references to sections and figures in the main text). Also, we believe our paper fits the scope of ACP (i.e., studies with general implications for atmospheric chemistry and physics) for the reasons mentioned above in our responses to AR1.

AR2: Please also highlight the novelty and differences with previous related studies in the Introduction.

We believe that we already highlight the novelty and differences with previous related studies in the introduction. We list previous papers that derive aerosol speciation or AMTs from satellite observations. We then proceed in describing what limits the derivation of AMTs from the POLDER satellite in one of our previous paper, Russell et al. 2014. We also specifically state how we expand on Russell et al. 2014 and the novelty of our method in this paper.

AR2: Many abbreviations need to give their full names where they first appear in the text, e.g., US EPA, SO2, NO2, O3, CO, … There are too many academic terms in the manuscript and it is suggested to add a table to summarize all acronyms and full names.

As previously mentioned in our responses to AR1, we have added a list of acronyms and abbreviations to this paper (Appendix B) and we have deleted some acronyms when they were found not essential.

AR2: The Figures can be improved, e.g., Figure 2: Better to present using the Flowchart, Figure 7: Add (a) to (c) for each subfigure, Figure 8: Add the legend for lines with different colors

Figure 2 was modified and is now in the format of a flowchart. Figure 7 now reads (a), (b) and (c) and Figure 8 now shows the legend for lines of different colors.

AR2: I suggest that the author can compare their results with satellite- and ground-based observations since there are aerosol optical properties and composition products and measurements available in the US.

To compare our in situ gas-phase, chemical or optical properties measured at the altitude of the aircraft to satellite and ground-based observations (we presume the reviewer refers to total column observations), we would have to first compare our in situ derived observations to on-board total column "effective" aerosol properties. This is discussed in section 5 of our paper (see "Another essential step should be to examine optical signatures from space-based passive remote sensor(s), which derive total column effective ambient aerosol optical properties (instead of properties measured at the altitude of the aircraft in this study)" and the following sentences). We believe that performing such analysis in this study would be outside the scope of our paper.